# Past and present giant viruses diversity explored through permafrost metagenomics

Sofia Rigou [1], Sébastien Santini[1], Chantal Abergel[1], Jean-Michel Claverie[1] & Matthieu Legendre [1] ✉

Giant viruses are abundant in aquatic environments and ecologically important through the metabolic reprogramming of their hosts. Less is known about giant viruses from soil even though two of them, belonging to two different viral families, were reactivated from 30,000-y-old permafrost samples. This suggests an untapped diversity of *Nucleocytoviricota* in this environment. Through permafrost metagenomics we reveal a unique diversity pattern and a high heterogeneity in the abundance of giant viruses, representing up to 12% of the sum of sequence coverage in one sample. *Pithoviridae* and *Orpheoviridae*-like viruses were the most important contributors. A complete 1.6 Mb *Pithoviridae*-like circular genome was also assembled from a 42,000-y-old sample. The annotation of the permafrost viral sequences revealed a patchwork of predicted functions amidst a larger reservoir of genes of unknown functions. Finally, the phylogenetic reconstructions not only revealed gene transfers between cells and viruses, but also between viruses from different families.

Permafrost, soil remaining continuously frozen for at least 2 years, covers 15% of the Northern hemisphere[1] and gathers complex communities of living organisms and variable soil types. The microbial community of the surface cryosol is in some cases subject to freezing and thawing of the soil every year[2] whereas communities from deeper layers are trapped as the sediments are deposited (syngenetic permafrost) or as the sediment freezes (epigenetic permafrost). Pleistocene permafrost has been shown to harbor up to $5 \times 10^7$ cells per wet gram of soil, a fifth of which are alive[3]. The permafrost has thus the ability to preserve organisms for tens if not hundreds of thousands years and acts as a huge reservoir of ancient microorganisms. For instance, it has been shown that numerous bacteria isolated from permafrost samples remained viable[4,5], even potentially for up to 1.1 million years[6]. Even in low biomass-containing frozen environments such as glacier ice, metagenomics approaches have recently revealed hundreds of distinct bacterial genera[7]. Unicellular[8–10] and even multicellular[11,12] eukaryotes can also be preserved for thousands of years and be revived from such frozen environments.

Besides cellular organisms, metagenomics studies have revealed bacteriophages communities archived in surface[13] or deeper[7] glacier ice, the majority of which being taxonomically unassigned. Due to the high bacterial abundance[14], bacteriophages are expected to be the most abundant viruses in the permafrost. However, in the unfiltered size fraction, the eukaryotic viruses *Nucleocytoviricota* (formerly known as Nucleocytoplasmic large DNA viruses or NCLDVs) are also highly represented[14]. This phylum gathers large double-stranded DNA viruses such as *Pokkesviricetes* (*Poxviridae* and *Asfarviridae*) as well as all the known giant viruses (i.e., viruses visible by light microscopy): the *Megaviricetes* (*Phycodnaviridae*, *Mimiviridae,* and *Pimascovirales*). More importantly, among *Nucleocytoviricota*, the two giant viruses *Pithovirus sibericum* and *Mollivirus sibericum*, were reactivated from a 30,000-y-old permafrost sample on *Acanthamoeba castellanii*[15,16]. Taking into account the presence of numerous protists (in particular ameba) in permafrost[9], many more giant viruses probably exist in such environments.

Recently, several studies specifically targeting environmental viruses have started to grasp the diversity and gene content of the *Nucleocytoviricota*[17–19]. They seem to be widespread in aquatic environments. More specifically, *Mimiviridae* (in particular the proposed *Mesomimivirinae* sub-family[20]) and *Phycodnaviridae* are major contributors of the marine viromes all over the world, as revealed by thousands of metagenome-assembled viral genome (MAG)

[1]Aix–Marseille University, Centre National de la Recherche Scientifique, Information Génomique & Structurale (Unité Mixte de Recherche 7256), Institut de Microbiologie de la Méditerranée (FR3479), 13288 Marseille Cedex 9, France. ✉e-mail: legendre@igs.cnrs-mrs.fr

sequences[17–19]. They have also been found active by metatranscriptomics at the surface layer of the ocean[21] and bloom-forming bays[22,23]. In addition to these two major groups, *Asfarviridae, Ascoviridae, Iridoviridae* and *Marseilleviridae* have been found active by marine metatranscriptomics[24]. Importantly, the genomes of giant viruses code for various auxiliary metabolic genes, making them capable of reprogramming their host's metabolism and thus, to potentially play an important role in global biogeochemical cycles[17,18,25].

The *Nucleocytoviricota* ecological functions and diversity in terrestrial samples are far less known, with the exception of *Klosneuvirinae* sequences recovered from forest soil samples[26] and of *Pithoviridae* sequences assembled from the Loki's castle deep sea sediments sequences[27]. The overwhelming proportion of *Nucleocytoviricota* metagenomic sequences of marine origin as compared to terrestrial is most likely due to the difficulty at revealing their hidden diversity in these environments[26]. Indeed, the high proportion of closely related strains in soil communities notoriously hampers sequence assembly, making soil metagenomic studies challenging[28,29].

Current giant viruses' metagenomic and metatranscriptomic studies rely on the detection of *Nucleocytoviricota* core genes[17,18,22,26,27]. However, among the handful of core genes, some of them are highly divergent or even completely absent from certain viral families. For instance, the Major Capsid Protein (MCP), often used as a marker gene to detect *Nucleocytoviricota* within metagenomic and metatranscriptomic assemblies[18,24], is absent from *Pandoraviridae*[30] and only present in a divergent form in *Pithoviridae*[15]. Thus, the probability to detect these types of non-icosahedral giant viruses is drastically lowered.

Although two distinct non-icosahedral giant viruses were initially isolated from permafrost samples[15,16], little is known about the diversity of *Nucleocytoviricota* in this type of environment. Here, we propose an analysis of these viruses from eleven permafrost samples ranging from the active layer up to 49,000-y-old sediment[14]. We show that the permafrost has a high viral diversity. Although the samples are very heterogeneous in *Nucleocytoviricota* content, they can reach up to an estimated relative abundance of 12% of the sequenced organisms (from sequence coverage). We found here that *Pithoviridae* and *Orpheoviridae*-like families followed by *Mimiviridae* are the main contributors of the giant virus diversity of the deep permafrost.

## Results

### Cryosol metagenomes assemblies

We gathered permafrost and surface cryosol raw metagenomic data from a previous study[14] on the three surface samples from Kamchatka (C-D-E, Supplementary Table 1) and eight deep samples from the Yukechi Alas area dated from 53 to over 49,000-y-old, four of which are syngenetic (Supplementary Table 1). Importantly, Cedratvirus kamchatka[31] and Mollivirus kamchatka[32] were isolated from the mentioned Kamchatka surface samples.

Previous analysis of this dataset[14] showed that prokaryotes are the most abundant (90% of the total coverage). Accordingly, the assembly of the reads (Supplementary Table 2) predominantly revealed bacterial contigs (mean = 94%, sd = 7%) according to the Lowest Common Ancestor (LCA) taxonomy based on BLASTP results against RefSeq. The samples with least bacterial contigs (N and R) still contain 80% of those, along with archaeal (10 and 7% respectively), unclassified (5%), viral (2%), and eucaryotic (1.3% and 1.6% respectively) contigs. Owing to the majority of bacterial contigs we reasoned that CheckM[33] could be applied to assess the overall validity of our assembly procedure of the complete dataset. This resulted in clean contigs with very few potential chimeras (0.004%) and no strain level chimera (Supplementary Fig. 1A). Next, as is custom in metagenomics studies, we performed a binning of the contigs to obtain less fragmented assemblies[34]. This revealed potential chimeras (Supplementary Fig. 1A). We thus chose not to consider bins as unique organisms but instead we used binning

as a procedure to decrease complexity in our datasets. More precisely, the reads were first separated according to the bin they belonged to. Next, a second de novo assembly was made within each bin. This resulted in significantly longer scaffolds and a larger total assembly (Supplementary Table 2) while keeping contamination at a negligible level (on average 0.005% potential chimeras and again none at the strain level, Supplementary Fig. 1A). Thus, our method significantly gained in reliability by lowering the proportion of chimeras in comparison to conventional binning, while providing longer assembled sequences compared to standard assemblies.

To further validate this strategy, we applied the same assembly method on three complex mock communities generated by a previous study[35]. Aligning the reference genomes used in that study on the resulting assembled sequences revealed a similar pattern: a clean first assembly, a noisier binned assembly, and a clean final assembly (Supplementary Fig. 1B). The proportion of chimeras in the final scaffolds accounts for only 0.2%.

### Discriminating *Nucleocytoviricota* in metagenomic samples

From the permafrost dataset we then sought to filter *Nucleocytoviricota* sequences. Our method is based on the detection of both *Nucleocytoviricota* genes (including the ones specific to the non-icosahedral *Pithoviridae* and *Pandoraviridae*) and cellular ones. We used a control metagenomic mimicking database containing reference *Nucleocytoviricota* genomes, cellular genomes randomly sampled from GenBank in addition to ameba and algae genomes (known hosts of *Nucleocytoviricota*) as well as ameba-hosted intracellular bacteria (*Babela massiliensis* and *Parachlamydia acanthamoebae*). Clearly the combination of the cellular and viral gene counts showed a very distinct pattern for *Nucleocytoviricota* compared to cellular genomic sequences (Fig. 1a). Using this control database, we computed the optimal parameters discriminating *Nucleocytoviricota* sequences (slope = 0.1, intercept = 1; Fig. 1), yielding high classification performance (sensitivity = 87.47% and specificity ≥99.53%; Supplementary Fig. 2). Taking proportions instead of viral and cellular ORFs counts did not yield better results (Supplementary Fig. 3). For comparison, we also tested the ViralRecall tool (35) that confirmed 1848 out of the 1973 (94%) scaffolds detected by our pipeline. Further controls for contamination in the *Nucleocytoviricota* dataset involved a search for ribosomal sequences, none of which were found. Manual functional annotation of all potential *Nucleocytoviricota* scaffolds allowed the identification of 7 scaffolds potentially belonging to intracellular bacteria, a phage and a nudivirus. All these sequences were removed. At the end, we identified 1966 *Nucleocytoviricota* scaffolds ranging from 10 kb up to 1.6 Mb in the permafrost dataset, corresponding to 1% of all scaffolds over 10 kb in size (Fig. 1b). Applying CheckM specifically fueled with viral HMM profiles made from low-copy NCVOGs (44) on this final sequence dataset resulted in virtually no contamination (mean = 0.0047%, s.d. = 0.027%) and strain heterogeneity (mean = 0.066%, s.d. = 1.8%).

As previously mentioned, *Nucleocytoviricota* metagenomic studies often rely on the MCP as a bait, making it hard, if not impossible, to catch some of the non-icosahedral viruses. By adding *Pithoviridae* and *Pandoraviridae* HMMs to the original profiles[18] and VOG's HMMs, we gained 5% (*n* = 110) more scaffolds that were mainly unclassified or from *Pithoviridae* and divergent *Pithoviridae* families (see further for phylogenies).

### Heterogeneous *Nucleocytoviricota* abundance in cryosols

The permafrost samples were very heterogeneous in *Nucleocytoviricota* relative abundance (Fig. 2) and number of scaffolds, ranging from 2 to 721 scaffolds, found in samples O (core permafrost under a lake in Yedoma, frozen for 40,000 years) and R (core permafrost under a drained thermokarst lake, frozen for over 42,000 years), respectively. *Nucleocytoviricota* scaffolds corresponded to 12% of the R sample

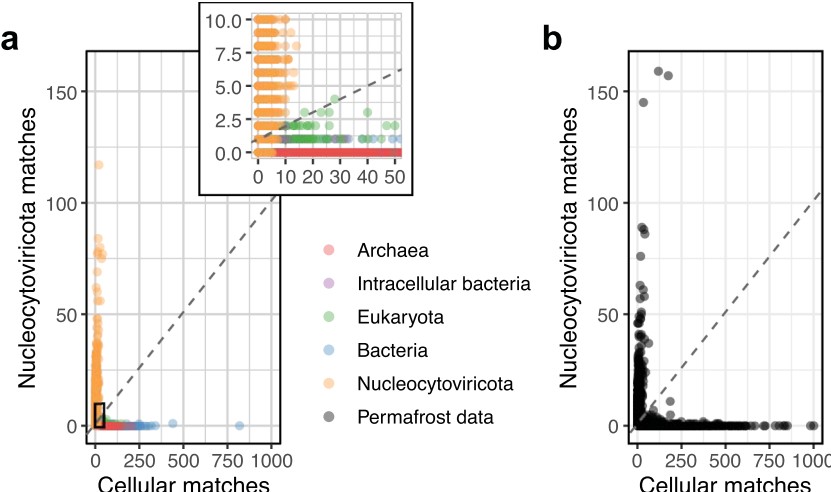

**Fig. 1 | Viral scaffolds filtering.** Each point corresponds to one scaffold. Viral matches (*y*-axis) were counted as the number of ORFs matching a *Nucleocytoviricota*-specific HMM at an *E* value of 10⁻¹⁰. These HMMs come from a previous study[18] to which were added specific HMMs from the VOG database and HMMs constructed on *Pandoraviridae* and *Pithoviridae* genomes. Cellular matches (*x*-axis) are the number of DIAMOND BLASTP matches against the cellular RefSeq database with a threshold of 35% of sequence identity and an *E* value ≤ 10⁻⁵. The dashed lines represent the chosen threshold excluding all point under or on the line. **a** Control dataset. The inset is a zoom of the bottom-left corner of the plot. For clarity, 1 bacterial point with over 1000 cellular matches and 1 viral match are not shown. **b** Permafrost data. For clarity, 5 points with over 1000 cellular matches are not shown. Source data are provided as a Source Data file.

sequence coverage (Fig. 2) and 17% of total reads mapped on scaffolds over 10 kb (4% of all raw reads). This sample was also the richest in eukaryotes with mostly Streptophyta (35%), Dikarya (14%), Platyhelminthes (9%), Eumycetozoa (8%) and Longamoebia (7%). Interestingly, amebas (Longamoebia) are on average 46.7 times more abundant in this sample than in the other ones (Supplementary Fig. 4A).

The relative proportion of giant viruses (Fig. 2) showed a strong correlation to the ones of Eukaryota. Precisely, Spearman correlation coefficients of $\rho = 0.72$ for the sum of coverages (two-sided correlation test *p* value = 0.017, Fig. 2) and $\rho = 0.83$ for the number of scaffolds (two-sided correlation test *p* value = 0.003) were observed. Such correlation could be explained by host-parasites dynamics. We therefore looked for potential co-occurrences of viral and eukaryotic families. Despite working with only 11 samples, we found significant associations (Supplementary Fig. 4B), including two *Pithoviridae*-like viruses with Entamoebidae. More surprisingly, we also found two other *Pithoviridae*-like associated with Hydrozoa. HGT between *Mimiviridae* and this eukaryotic class has already been observed[36]. Finally, two other *Pithoviridae*-like were also found associated with Cryptomonadaceae. Although these eukaryotes are not known to be infected with giant viruses, metagenomics co-occurrence analyses showed association between cryptophytes and *Mimiviridae*[22] as well as virophages[37].

*Nucleocytoviricota* scaffolds could also correspond to endogenized viruses in eukaryotes (GEVE), as previously shown in green algae[38]. This hypothesis is plausible as 57% (193 out of 338) of the GEVE pseudo-contigs (see Methods) were captured by our *Nucleocytoviricota* detection method. To explore this possibility, we thus checked for endogenization signs in the viral scaffolds using ViralRecall[39] (example in Supplementary Fig. 5) but none was found. In addition, *Nucleocytoviricota* largely outnumber eukaryotes with a 4:1 *Nucleocytoviricota*/Eukaryota ratio in the sum of coverages (mean = 4.06, s.d. = 4.22) and number of scaffolds (mean = 4.40, s.d. = 3.34). Altogether, this suggests that most of the discovered permafrost *Nucleocytoviricota* scaffolds correspond to bona fide unintegrated viruses.

### Exploration of the sequence diversity
To further investigate which viral families were present in the samples, we conducted a phylogenetic analysis based on 7 marker genes (Supplementary Data 1) and a curated database produced by a former study[40]. We excluded the transcription elongation factor TFIIS as its phylogeny breaks well-established clades (*Alphairidovirinae*, *Ascoviridae*, *Asfarviridae*, *Pimascovirales*, Supplementary Fig. 6). It should also be noted that the primase D5 revealed an unexpected grouping of the Cedratviruses with *Phycodnaviridae* instead of *Pithoviridae*, suggesting that this gene was acquired from an unknown source in Cedratviruses (Supplementary Fig. 6). We first classified permafrost scaffolds containing at least three of the seven marker genes to avoid split genomes in the tree. This resulted in 37 classified scaffolds (corresponding to 16.5% of the 72 Mb of total *Nucleocytoviricota* identified sequences) with 21 scaffolds within the *Pithoviridae* and *Orpheoviridae*-like clades, 8 in the *Megamimiviridae* clade and the rest associated to *Klosneuviridae*, *Phycodnaviridae* and one *Asfarviridae* (Fig. 3a).

However, filtering scaffolds with less than three marker genes only reveals the ones representing a substantial portion of the viral genome and thus probably under-estimate the true diversity of viral families. Indeed, counts derived from single markers (Fig. 3b) show that *Pithoviridae* and *Orpheoviridae*-like sequences might be particularly under-estimated as they lack the packaging ATPase and contain a highly divergent MCP. In addition, they contain a substantially lower fraction of duplicated marker genes than *Megamimivirinae* and *Klosneuvirinae* (Fig. 3b). We thus also performed a classification of all scaffolds containing at least one marker gene. This increased the taxonomically classified dataset to 369 *Nucleocytoviricota* scaffolds (40.1% of the *Nucleocytoviricota* sequences). Again, *Pithoviridae* and *Orpheoviridae*-like viral families were the most diverse, followed by *Mimiviridae* (Supplementary Fig. 7). In contrast, *Marseilleviridae*, *Alphairidovirinae*, *Betairidovirinae,* and *Ascoviridae* were completely absent in our samples. Interestingly, unclassified sequences do not encode for more ORFans (ORFs with no similar sequence in the public databases) than classified sequences (Supplementary Fig. 8A). This suggests that these sequences are not more divergent to known relatives than any other *Nucleocytoviricota* sequence but remained unclassified due to the lack of the marker genes.

We further confirmed the observed taxonomy pattern from individual marker genes phylogenies (Supplementary Fig. 9) and the best BLASTP matches of the unclassified sequences against the nr database (Supplementary Fig. 8B). Finally, an alternative phylogeny of the bins (instead of scaffolds) probably noisier but representing 85.4%

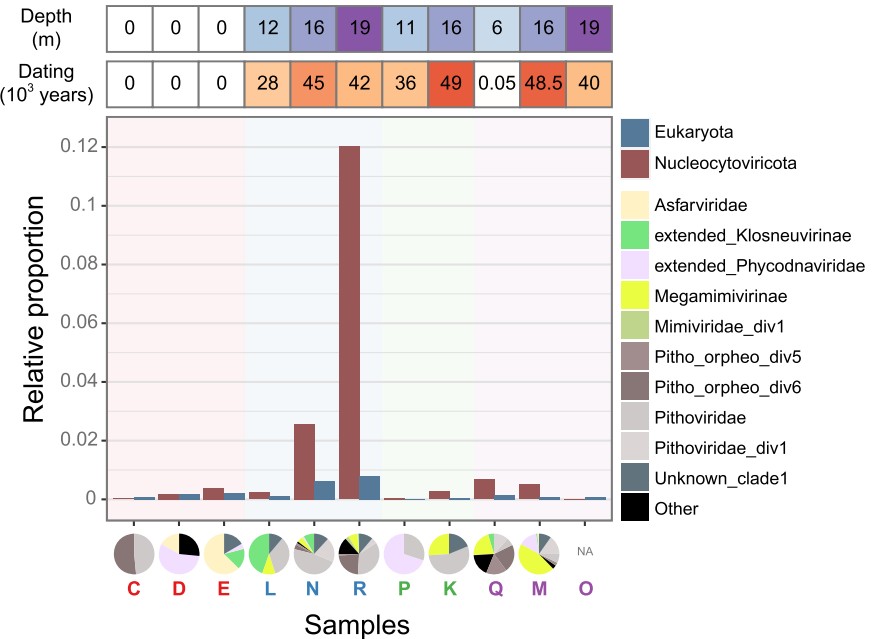

**Fig. 2 | Relative abundance of *Nucleocytoviricota* and Eukaryota across samples.** The relative abundance is calculated as the sum of scaffold coverages belonging to the given group divided by the total sample coverage among scaffolds ≥10 kb. Sample names in red are surface samples from Kamchatka while samples in blue, green and purple indicate that they come from three different forages in the Yukechi Alas area. The pie charts indicate the taxonomy of the *Nucleocytoviricota* in different samples (see further for phylogeny) whose abundance has also been estimated from the scaffold coverages. Only classified scaffolds were considered. Source data are provided as a Source Data file.

of the total *Nucleocytoviricota* sequences confirms the pattern (Supplementary Fig. 10). Altogether, these results clearly support *Pithoviridae* and *Orpheoviridae*-like as the most diverse families in our samples.

Most viruses are specific to the sample they were recovered from, in particular the ones from surface samples (Supplementary Fig. 11). Surprisingly, we also found viruses that were common to samples from close locations in Central Yakutia but from different ages (samples K, L, M, N, P, Q, and R; Supplementary Table 1). As the samples are unlikely contaminated[14], this indicates that part of the viral community was maintained over time.

**Enrichment of *Pithoviridae* and *Orpheoviridae*-like genomes in the Permafrost**

Not only *Pithoviridae* were unexpectedly diverse (Fig. 3, Supplementary Figs. 7 and 10), they were also the most abundant *Nucleocytoviricota* according to their normalized coverage (Fig. 2). *Pithoviridae/Orpheoviridae*-like families appear in all samples and are particularly abundant in samples R and N (Fig. 2). The single most covered (i.e., abundant) sequences in five samples (C, N, R, K, and Q) come from these, and from Extended_phycodnaviridae, *Megamimivirinae* and *Klosneuvirinae* in the other samples.

The *Pithoviridae* diversity and abundance observed in the Siberian permafrost (in particular in samples R and N) could either highlight the enrichment of this viral family in this environment or represent the improvement in our method for the detection of non-icosahedral viruses. To compare the diversity found in the presented samples to other soil environments, we applied the same detection method to 1835 terrestrial datasets collected from the JGI IMG/M database[41]. The vast majority of these terrestrial samples exhibited no *Nucleocytoviricota* sequences and few contigs over 10 kb in general, probably due to the difficulty at assembling sequence data from these complex environments. Comparatively, the diversity of *Pithoviridae* and *Orpheoviridae* observed in the cryosol samples is unique as they were significantly enriched in these viruses, followed by forest soil (Supplementary Fig. 12). Noteworthy, Pandoravirus-like sequences were found in sand and a 900 kb contig grouping next to *Pandoraviridae* and *Molliviridae* in peat permafrost samples.

**Large viral genome fragments from the deep permafrost**

Although our strategy to exclude conventional binning was primarily designed to capture high confidence MAGs at the price of completeness, we were still able to recover large *Nucleocytoviricota* genomes in single scaffolds with no apparent chimera (see "Methods"). Eight of them, assembled from 16 m to 19 m deep permafrost samples (R, N and M, Supplementary Table 1) dating from 42,000 to 49,000 years, reached over 500 kb (Fig. 4). The largest one of 1.6 Mb, referred to as "Hydrivirus", is likely complete as it was successfully circularized. Although these large scaffolds are deeply sequenced (with an average coverage in between 14 and 72), they do not belong to the most abundant viruses in their samples (the highest coverages are of 53, 181, and 1572 in samples M, N, R respectively).

These MAGs vary in divergence from known genomes, having from 22% up to 72% of ORFans for Unknown Permafrost:M_b2437_k1 (Fig. 4). As is common for newly discovered giant viruses, their genomes also match cellular genes from all domains of life (with very few Archaea). The four largest scaffolds were classified within *Pithoviridae/Orpheoviridae*-like families. Two are putative *Megamimivirinae* (Mimivirus Permafrost:R_b548_k1 and Mimivirus Permafrost:R_b2349_k1) and one is a putative *Klosneuvirinae*. Finally, the Unknown Permafrost:M_b2437_k1 scaffold is placed near the root of the tree (Fig. 3) and its evenly distributed viral best BLASTP matches have no specific family standing out (Fig. 4). Taking its scaffold phylogeny (Fig. 3 and Supplementary Fig. 7) together with its high ORFan content suggests that it belongs to a *Nucleocytoviricota* viral family with no isolate so far.

The complete 1.6 Mb Hydrivirus genome reaches a size similar to the isolated Orpheovirus[42]. The other 715 to 855 kb scaffolds are slightly larger than isolated *Pithoviridae* (ca. 600 kb)[15,43,44]. However, they were not circularized as expected for a *Pithoviridae* genome structure[15] and are thus potentially even larger. Still, in the four of them, most of the core genes are present (Supplementary Data 1).

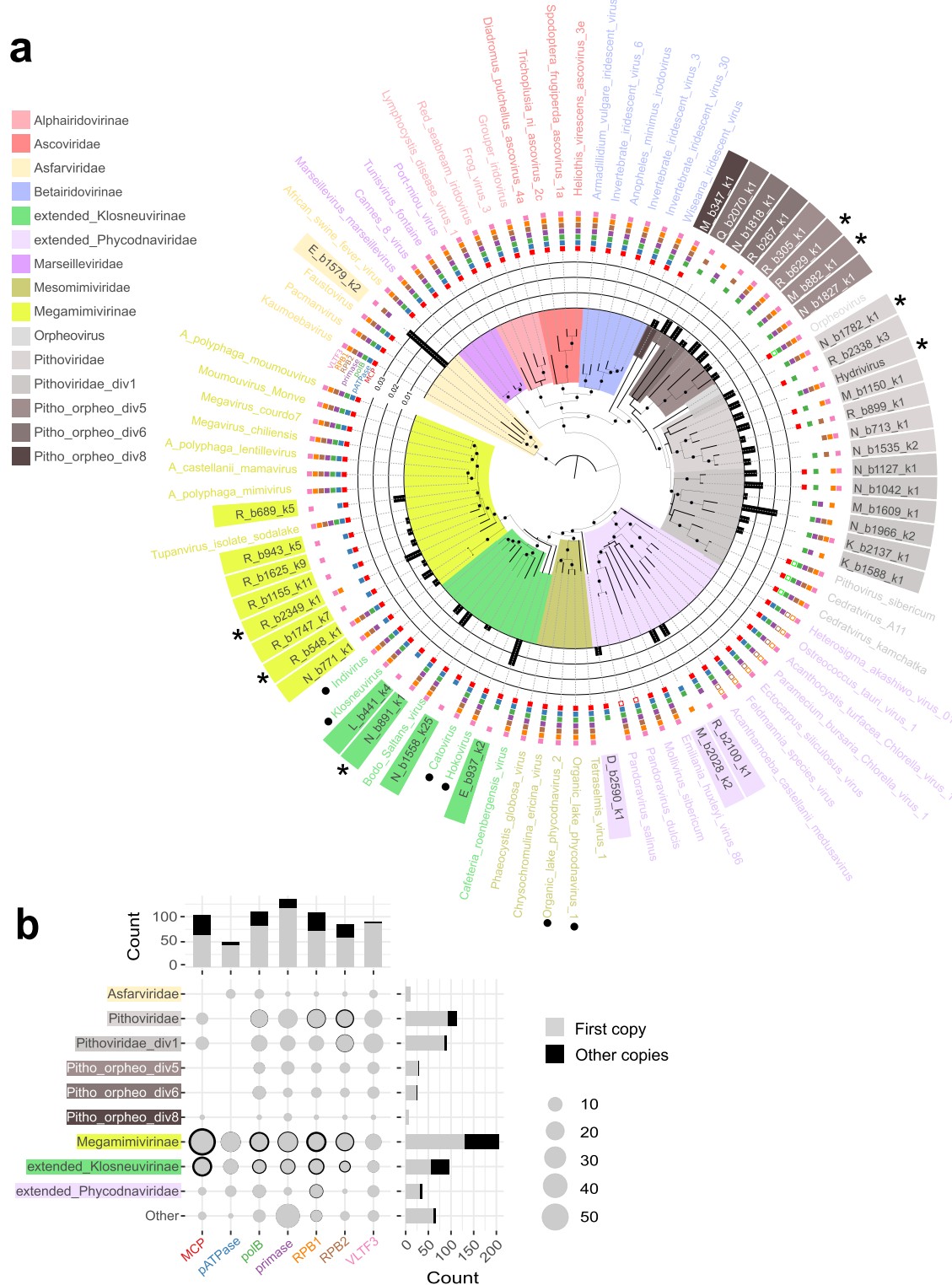

**Fig. 3 | *Nucleocytoviricota* diversity in all samples. a** Consensus of 1000 bootstrapped trees calculated by IQ-TREE through a partitioned analysis on 7 marker genes. The tree was performed on permafrost sequences (colored background labels) with more than three marker genes and on reference sequences (white background). The stars indicate the large (>500 kb) scaffolds identified in this study. The models used were the following: LG + F + I + G4 for the packaging ATPase, the RNA polymerase subunits, VT + F + I + G4 for the DNA polymerase, VT + F + G4 for VLTF3 and LG + F + G4 for the MCP. Black dots represent branch bootstrap support ≥ 90%. One should note that reference genomes coming from bins of previous metagenomic studies (marked with a black dot) are less reliable than the genomes of isolated viruses. The colored clades were manually created to be monophyletic. The marker genes used for this phylogeny are indicated as colored squares. Empty squares correspond to marker genes absent from the reference genomes. Black bars show the relative mean coverage of the scaffolds (%). The Extended_phycodnaviridae group includes *Pandoraviridae* and Mollivirus. The Extended_klosneuvirinae group includes the Cafeteria roenbergensis virus. **b** Total marker gene count associated to the taxonomy of scaffolds with at least one marker gene. Total counts of each viral clade and each marker gene are shown as barplots on the right and top panels, respectively. Source data are provided as a Source Data file.

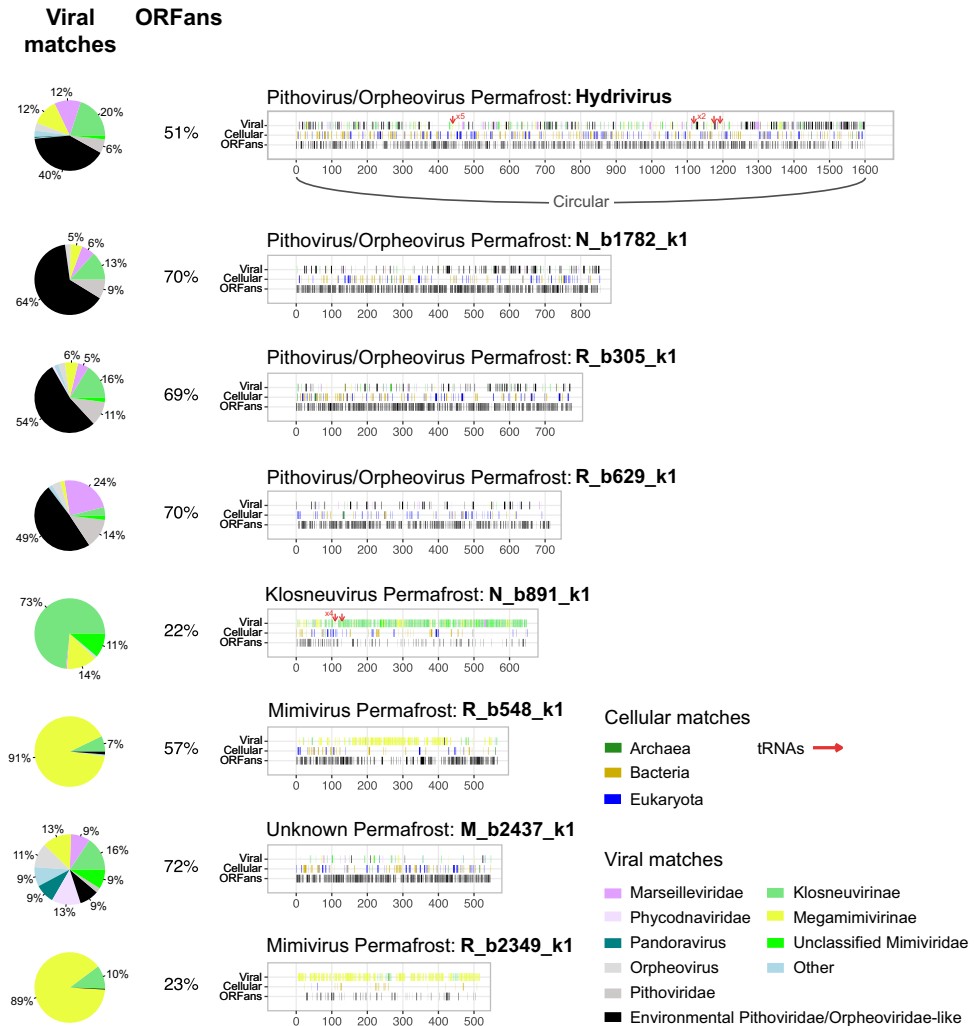

**Fig. 4 | Gene content of the large genomes recovered from ancient permafrost samples.** For each genome, the position of ORFans (ORFs with no match in the nr database), cellular and viral matches are recorded along the genome. The positions of tRNAs are also shown as red arrows. The pie charts present the proportion and taxonomy of viral matches with slices ≥5% labeled. Groups that match less than 5% of the Unknown Permafrost:M_b2437_k1 scaffold were gathered in "other" except for *Pithoviridae*. The environmental *Pithoviridae/Orpheoviridae*-like category contains metagenomic sequences from Bäckström et al.[26, 27]. The Hydrivirus genome was circularized. Source data are provided as a Source Data file.

Furthermore, except for Pithovirus/Orpheovirus Permafrost:R_b629_k1, all the *Pithoviridae*-like large genomes and Klosneuvirus Permafrost:N_b891_k1 have a near complete base excision repair system.

## Functions encoded in the permafrost *Nucleocytoviricota* sequences

To get insight into the functions encoded by the permafrost *Nucleocytoviricota* we manually annotated a total of 64,648 viral ORFs over 50 amino acids that were assigned to functional categories. The distribution follows the one of *Nucleocytoviricota* references (Supplementary Fig. 13), with most of the predicted proteins (81%) being of unknown function (as compared to 64% in reference genomes, Supplementary Fig. 13). We searched for significantly enriched Pfam and Gene Ontology annotations in the permafrost viral datasets compared to references but found none after false discovery p-value correction apart from a couple of core function (Supplementary Data 2). We also did not find specific functional enrichment when comparing samples to each other within the same viral families (Supplementary Data 3). Likewise, when mixing all viral families together, ecological parameters do not discriminate samples based on Pfam annotations (Supplementary Fig. 14). Altogether, this indicates that viral genome content

and ecological parameters are not directly correlated or, more likely, that the high proportion of genes with unknown functions and the limited number of samples prevent this from being revealed at this time.

As expected from their reference counterpart, permafrost *Nucleocytoviricota* encode auxiliary metabolic genes that are scattered within the different viral families (Supplementary Figs. 15 and 16). In addition, they encode functions not previously observed, such as ATP synthases subunit F (in the N_b713_k2 Pithoviridae_div1 sequence), as well as truncated hemoglobins in 3 permafrost *Pithoviridae*-like (R_b2567_k1, M_b1150_k2 and N_b1127_k2) and in an Extended_phycodnaviridae sequence (M_b2028).

Looking at highly shared functions (i.e., present in most families) among the reference genomes and permafrost MAGs, we identified the known core genes (Fig. 5) with the exception of the mRNA capping enzyme, absent from the *Iridoviridae/Ascoviridae* clade. The patatin phospholipase, suspected to be conserved among *Nucleocytoviricota*[45], is confirmed as a core gene, only absent from *Alphairidoviridae* (Fig. 5). Conversely the A32-like packaging ATPase presumably encoded by a "core" gene in large DNA viruses is no longer a universal *Nucleocytoviricota* marker gene, as it is not only lacking from the reference *Pithoviridae* genomes[46] but also absent from all

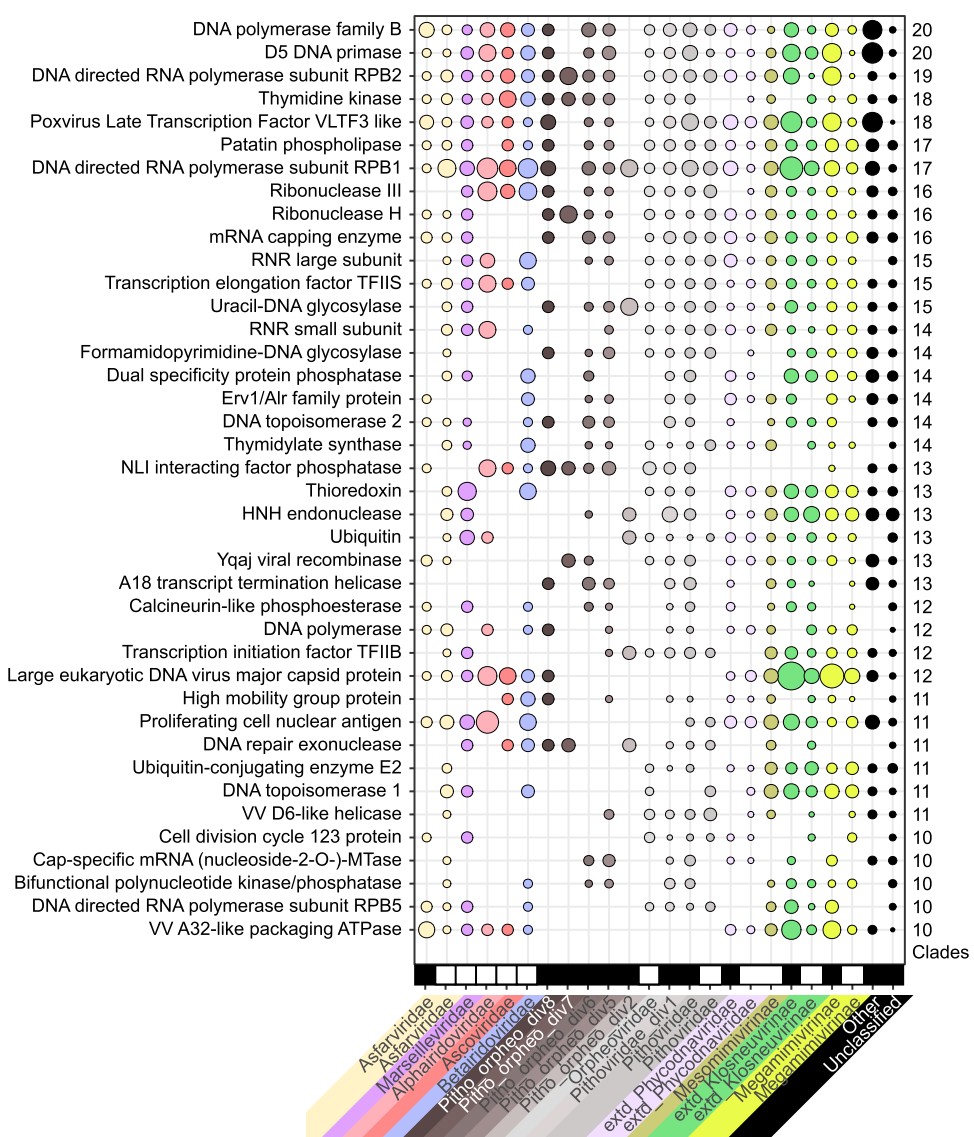

**Fig. 5 | Comparison of most shared functions among metagenomic and reference *Nucleocytoviricota* families.** Functions were selected among the annotations found in at least 10 clades. Metagenomic sequences are marked as black rectangles at the bottom of the plot while blank spaces correspond to reference genomes. Groups with less than 250 ORFs were marked as "Other". The size of the bubbles represents the normalized ORFs counts (i.e., ORF counts/total number of ORFs in the group). The right-most column indicates the number of distinct clades having the function. The lines are sorted according to this value. Source data are provided as a Source Data file.

clades ranging from Pitho-orpheo_div8 to *Pithoviridae* (Fig. 5). Overall, our analysis highlights a patchwork of functions encoded by these viruses (Supplementary Figs. 15 and 16).

**Virally-encoded translation-related genes**

Virally-encoded translation-related genes are a landmark of giant viruses. They were found in a large spectrum of viral families and might give clues on their evolution and interaction with cellular organisms. We thus specifically analyzed the translation-related genes in the permafrost data and found 20 different types of virally-encoded aminoacyl-tRNA synthetases (aaRSs). As previously observed in other *Klosneuvirinae*[47], the Klosneuvirus Permafrost:N_b891_k large genome fragment (Fig. 4) encodes an expanded translation-related gene repertoire (10 translation initiation factors, 4 translation elongation factors, a translation termination factor, 11 different aaRSs and 5 tRNAs clustered together). More surprisingly, ten different types of aaRSs were also found in the Pithoviridae_div1 clade, including 7 different ones in the Hydrivirus genome (Fig. 4). This virus also encodes 9 tRNA,

3 translation initiation and elongation factors, and a translation termination factor.

We then investigated the phylogeny of the different types of aaRSs found in our datasets that revealed entangled evolutionary pathways between viruses and cellular organisms (Supplementary Figs. 17–19). In most cases, the viral aaRSs were likely acquired by HGT from Eukaryotes (tryptophan, leucine, glutamine, threonine, methionine, isoleucine, arginine, aspartate, serine and phenylalanine) (Supplementary Figs. 17 and 19). In rare cases, we detected a possible HGT from an Archaea to a virus as for the glycine- and tyrosine-tRNA synthetases (Supplementary Fig. 18). Genes have also passed from Bacteria to *Nucleocytoviricota*, as for the glycine-tRNA synthetase of Hydrivirus and the valine-tRNA synthetase of a permafrost *Megamimivirinae*. For the latter, the bacterial sources were Rickettsiales, which are endosymbionts of ameba[48], and thus probably share the same host. The source of the tryptophan-tRNA synthetase in Hydrivirus is less clear, but a duplication event occurred probably at the same locus right after the gene was acquired (Supplementary Fig. 17).

While the vast majority of *Nucleocytoviricota* genes have no identifiable homologs, the ones with cellular homologs usually deeply branch in the phylogenetic trees[17,49], in accordance with their suspected ancient origin[40,50]. We found here several viral aaRSs that belong to divergent families tightly clustered together within the cellular homologs (Supplementary Fig. 19). So not only viral aaRSs are of cellular origin, spanning all domains of life, they were also probably exchanged between viruses of different families.

## Discussion

Recent large-scale metagenomic data analyses strikingly revealed that *Nucleocytoviricota* are widespread in various environments[17,18,26,27]. Our analysis of cryosol samples confirmed this ubiquity. Nevertheless, we highlighted an important heterogeneity in *Nucleocytoviricota* proportions across the samples, in agreement with the already observed heterogeneity at various scales (domain, phylum, class, order and functional annotation) for all domains of life[14]. This heterogeneity is probably the testimony of the absence of mixing between layers but also of a spatially heterogeneous microbiome. It has been pointed out that the bacterial community of agricultural soil changes at a centimeter-level[51]. Thus, the heterogeneity we observe might translate a sampling bias although probably attenuated by the large amount of soil from each sample (20 g) used for DNA extraction. In any case, such heterogeneity includes eukaryotes which likely strongly influences the abundance of *Nucleocytoviricota*. The co-occurrence analysis of *Nucleocytoviricota* and eukaryotes performed in this study linked *Pithoviridae*-like clades to ameba. More surprisingly others were associated with Cryptomonadaceae and Hydrozoa. This widens the possible host range of *Pithoviridae* in the same way *Mimiviridae* infect various distant clades[52]. However, co-occurrence might translate indirect correlation and not direct virus-host interactions.

Importantly, our samples are among the most enriched in *Nucleocytoviricota*, reaching up to 12% of the estimated abundance of sequenced organisms. Moreover, the relative DNA sequence coverage (Fig. 2) suggests that they outnumber their hosts, in the same way bacteriophages often outnumber bacteria in the ocean[52,53]. This high abundance is the result of a high diversity in the samples, as it does not come from a single virus that would be at the origin of all sequenced reads since the individual maximum relative coverage never exceeds 0.3%. Furthermore, by taking advantage of the permafrost's ability to preserve ancient organisms, we showed that some *Nucleocytoviricota* strains have been present in the surface community for a long time (Supplementary Fig. 11). Considering only syngenetic permafrost samples, we found *Nucleocytoviricota* shared in samples of up to 13,000 years difference. This indicates that they probably are important players of this particular area of central Yakutia.

The *Nucleocytoviricota* diversity explored in this study revealed large genomic sequences of unknown families, such as the Permafrost:M_b2437_k1 scaffold (Fig. 4). In addition, we identified many divergent *Pithoviridae*-like sequences which constitute new clades within the *Pimascovirales*. In contrast, *Megamimivirinae*, *Klosneuvirinae* and *Mesomimivirinae* were the groups with the least ORFans within the permafrost sequences. These groups are thus better sampled than all other viral families found in this study (Supplementary Fig. 8A). Overall, the high ORFan content in our dataset probably explains the paucity of functions significantly enriched in permafrost samples compared to reference sequences. In addition, the patchwork-like pattern of *Nucleocytoviricota* functions might also blur the statistical signal.

Importantly, the 1.6 Mb Hydrivirus genome recovered by our method is complete. So, together with Pandoraviruses[30], Orpheoviruses[42], Klosneuviruses[47], and Mimiviruses[54], Hydrivirus is another example of a viral genome largely over 1 Mb. The nature of the evolutionary forces pushing some viruses to retain or acquire so many genes remains a matter of debate[55–58]. Horizontal gene transfers from

cellular hosts is hypothesized by some authors to account for their large gene content[47,59]. We indeed found examples of cellular genes gained by HGT in this study (Supplementary Figs. 17–19) but this only accounts for a small proportion of their gene content, with the vast majority having no identifiable cellular homolog. Gene duplication, on the other hand, a well-known source of functional innovation since the pioneering work of Susumu Ohno[60], may contribute to the genome inflation of giant viruses[49,61]. Another possible source of genetic innovation is the de novo gene creation from intergenic regions[49,62]. The present work expanded the *Nucleocytoviricota* families' pangenomes, in particular the *Pithoviridae*-like with an overwhelming proportion of ORFans. Part of these genes might originate from de novo gene creation, a hypothesis that remains to be further tested.

The functional annotation performed in this work highlights the paucity of functions strictly shared between *Nucleocytoviricota*. This includes proteins thought to be central for viral replication/transmission, like the A32 Packaging ATPase[46] which is absent from the entire *Pithoviridae*-like clade (Fig. 5). Likewise, the MCP is not encoded in the *Pandoraviridae* genomes[30]. Our work also highlights a patchwork of functions and independent cases of HGT from Eukaryotes to viruses but also between viruses belonging to different families (Supplementary Figs. 17–19). This is probably the testimony of coinfections, as members of the *Marseilleviridae*, *Mimiviridae*, *Pithoviridae*, *Pandoraviridae* and *Molliviridae* families can infect the same host. Although endogenization may blur the counts, it was estimated by single-cell sequencing that as much as 37% of cells carry 2 or more viruses[63], thus promoting gene exchanges between viruses. In line with this hypothesis, it was recently showed that DNA methylation, widespread in giant viruses, is mediated by methyltransferases and Restriction-Modification systems that are frequently horizontally exchanged between viruses from different families[31].

The functional patchwork, the gene exchanges between viruses of different families and the very few shared genes may challenge the monophyly of the recently established *Nucleocytoviricota* phylum by the International Committee on Taxonomy of Viruses (ICTV)[64]. Except for the DNA primase of Cedratviruses, our trees of seven marker genes would indeed indicate a shared ancestry of the different *Nucleocytoviricota* families analyzed in this work (Supplementary Fig. 6). However, when cellular genes are integrated to the phylogenetic trees, only three of the five most shared genes strictly support the monophyly of the *Nucleocytoviricota*[65]. These are the viral late transcription factor 3, the Holliday junction resolvase and the A32 packaging ATPase genes. The latter has also been shown to be exchanged between *Mimiviridae* and Yaravirus, an Acanthamoeba infecting virus that does not belong to the phylum[65,66]. The other core genes such as the DNA polymerase is separated by several cellular clades between *Pokkesviricetes* and *Megaviricetes*[67]. Likewise the two largest subunits of the RNA polymerase of *Asfarviridae* and *Mimiviridae* have a different history than the other *Nucleocytoviricota*[40]. These examples question the consistency of the phylum.

The objective of this study was to assess the diversity of large DNA viruses in permafrost. Our analyses revealed an unexpected number of unknown viral sub-groups and clades, some among of the previously established families of the *Nucleocytoviricota* phylum. The phylogenetic diversity recovered from the ancient permafrost translated into an intricate functional patchwork amidst a majority of anonymous genes of unknown functions.

## Methods

No approval by board/committee and institution was required for this study.

### Data preparation

Illumina sequencing reads from all samples (Supplementary Table 1) were assembled into contigs using MEGAHIT (v1.1.3)[68] and then binned

using Metabat2[69] (v2.15) with a minimal contig length of 1500 and bin length of 10,000. Reads corresponding to each contig were retrieved and gathered from their respective bins using an in-house script. The read subsets were then reassembled using SPAdes[70] (v3.14) in default mode or with the "–meta" option. Reads were mapped on the resulting scaffolds ≥10 kb using Bowtie 2[71] (v2.3.4.1) with the "–very-sensitive" option and filtered with SAMtools (-q 3 option). Reads ≤30 nucleotides were discarded. Scaffold relative abundance was estimated as the mean scaffold coverage divided by the total sample coverage. Bins, contigs and scaffolds were verified with CheckM[33] (v1.1.2) using the lineage workflow. CheckM was also applied on a custom set of HMMs made from the NCVOGs database[45] using the "analyze" and "qa" tools. NCVOGs with 1.1 copies or less in average were used to construct the HMM profiles for a low-copy NCVOG database.

The validation of the method was performed on three datasets from a previous study[35]. We used high complexity mock communities with strain diversity within each species (ani100_cHIGH_stTrue_r0, ani100_cHIGH_stTrue_r1 and ani100_cHIGH_stTrue_r2) on which we applied the same assembly procedure (see previous paragraph). We then aligned the resulting contigs and scaffolds to the corresponding reference genomes with BLASTN (from BLAST + v2.8.1, options -evalue 1e-10 -perc_identity 99)[72]. Then matches ≥99.99% of identity were cut if overlapping with previous better matches and kept if they were ≥500 nucleotides using an in-house script. Bins with only one contig were not considered to assess the level of chimerism of bins or of the second assembly. With these data we assessed the proportion of chimeras (a contig, bin or scaffold matching different genomes) at each assembly step.

### Control database preparation

Reference *Nucleocytoviricota* were chosen following a former phylogenetic study[40]. The corresponding genomes were gathered from the NCBI repository. Lausannevirus, Melbournevirus, Ambystoma tigrinum virus, Infectious spleen and kidney necrosis virus, Invertebrate iridovirus 22, Invertebrate iridovirus 25 and Singapore grouper iridovirus were removed to avoid an overrepresentation of their families. We added the genomes of *A. castellanii* medusavirus (AP018495.1), Bodo saltans virus (MF782455.1), Cedratvirus kamchatka (MN873693.1) and Tetraselmis virus 1 (KY322437.1). Genomes from Archaea, Eukaryota, and Bacteria (Supplementary Data 4) were retrieved from GenBank. For each genome, non-overlapping sequences were cut with an in-house script following a distribution similar to our dataset to simulate metagenomic contigs. Genes were then predicted by GeneMark (v3.36)[73] using the metagenomic model. For the *Nucleocytoviricota* phylogeny, core genes previously identified[40] were used in addition to the ones found by PSI-BLAST[74]. We also added Amsacta moorei entomopoxvirus (AF250284.1), Variola virus (NC_001611.1) and Cyprinid herpesvirus 2 (MN201961.1) as outgroup.

### *Nucleocytoviricota* specific profiles databases

The database constructed by Schulz et al.[18] was completed with specific signatures of *Pithoviridae* using the genomes of Cedratvirus A11[44], Cedratvirus kamchatka[31], Cedratvirus lausannensis[75], Cedratvirus zaza[76], Brazilian cedratvirus[76], Pithovirus massiliensis[43], Pithovirus sibericum[15], Orpheovirus[42], all the metagenomic *Pithoviridae* (with the exception of Pithovirus LCPAC101) released from one study of Loki's Castle hydrothermal vents[27], the divergent *Orpheoviridae/Pithoviridae* SRX247688.42[17], the GVMAG-S-1056828-40[18] and other Cedratvirus/Pithovirus sequences (Supplementary Data 5). For *Pandoraviridae* we gathered sequences from Pandoravirus braziliensis[77], Pandoravirus celtis[62], Pandoravirus dulcis[30], Pandoravirus inopinatum[78], Pandoravirus macleodensis[49], Pandoravirus neocaledonia[49], Pandoravirus pampulha[77], Pandoravirus quercus[62], Pandoravirus salinus[30], Mollivirus kamchatka[32] and Mollivirus sibericum[16]. The ORFs were then predicted using GeneMark (v4.32) with the "–virus" option and ORFs ≥50 amino

acids were kept. Orthogroups were calculated with OrthoFinder[79] and HMM profiles were built using the Hmmer suite[80] (v3.2.1) for each one. HMMs were further aligned to the RefSeq protein database (from March 2020) using the same suite. Only HMMs specific to *Pithoviridae*, *Orpheoviridae*, *Pandoraviridae* or Molliviruses with $E$ value ≤ $10^{-10}$ were kept to complete the database. To these were added *Nucleocytoviricota*-specific VOG orthogroups (https://vogdb.org/).

### Retrieving viral sequences

The *Nucleocytoviricota*-specific profile database was searched against the control and permafrost ORFs using hmmsearch (with $E$ value <$10^{-10}$). To check for cellular signatures, all the ORFs were aligned to the RefSeq protein database using DIAMOND BLASTP (v0.9.31.132) with the "–taxonlist 2,2759,2157" option and hits ≥ 35% sequence identity and $E$ value <$10^{-5}$. On the control metagenomic simulated dataset, the number of false positives and false negatives were assessed according to the cellular and viral matches for each group (*Nucleocytoviricota*, Archaea, Bacteria, Eukaryota). We set the threshold at less than 1% of false eukaryotic positives. The same threshold was applied to the permafrost data to retrieve viral scaffolds. For comparison, we also tested ViralRecall (v2.0) with the "–contiglevel" option, contigs with a score >0 were considered as viral.

### Functional annotation

All the ORFs ≥ 50 amino acids were queried against the nr database (from June 2020) using BLASTP, the VOG database using hmmsearch, the Pfam database using InterProScan (v.5.39-77) and against EggNOG[81] using the online version of Emapper-1.03. For all, the $E$ value threshold was set to $10^{-5}$. Functional annotations of each predicted protein were defined manually, first based on the matching domains annotations, then by considering the full sequence alignments (BLAST, EggNOG and VOG). EggNOG categories were also set manually for each gene. When existing, the functional annotations of reference viral genomes (see control database preparation) were retrieved from GenBank. Grouper iridovirus, Heliothis virescens ascovirus 3e and Invertebrate iridescent virus 6 were manually reannotated using the same protocol as for the permafrost ORFs.

### Functional enrichment analysis

The Pfam and GO term annotations were retrieved from the InterProScan output for statistical analysis. Each Pfam annotation was compared either to the references or between samples within taxonomical groups using fisher exact tests to search for enriched functions. The p-values were corrected for multiple testing using the Benjamini & Hochberg FDR correction. Biological Processes GO terms were analyzed using the topGO package with the "weight" algorithm. Samples were also compared to each other based on viral Pfam annotations with all viral families together applying Bray-Curtis dissimilarity for clustering. We used the nonpareil diversity of the complete sequence data of each sample computed in Rigou et al.[14]. Lastly, in order to search for complete or near complete functional pathways in the large genome fragments recovered we screened them for KEGG annotations using BlastKOALA[82].

### Contamination control

The functional annotation step helped to remove non-*Nucleocytoviricota* scaffolds based on the presence of typical viral/phage genes or with ORFs consistently matching cellular organisms. The scaffolds were checked for the presence of ribosomes using Barrnap (v0.9)[83]. Finally, we checked for possible GEVEs (Giant Endogenous Viral Elements) in our curated scaffolds. We made pseudo-contigs from the GEVEs identified by Moniruzzaman et al.[38] and applied our method on them. As 57% (193 out of 338) of the GEVEs peudo-contigs were caught, we proceeded to check for endogenization signs in our permafrost scaffolds. This was done by plotting the domain of the best BLASTP

hits as well as the VOG matches for each scaffold with the results of the ViralRecall (v2.0) rolling score using default parameters[39]. Scaffolds with at least one region with a negative ViralRecall score were visually inspected.

### Large genomes assembly verification and circularization
The eight largest MAGs (≥500 kb) were scrutinized for possible chimeric assemblies. First, we checked visually that there was a single trend in the coverage along the scaffolds in log scale. Then we used the Integrative Genome Viewer[84] to scrutinize the positions where the coverage dropped under 3× (mainly due to ambiguous bases added during scaffolding). In each case, read pairs overlapped the low coverage intervals. For circularization, we created a model contig concatenating both ends of the MAG, mapped the reads using Bowtie 2 and visually checked the uniformity of the coverage at the junctions using the Integrative Genome Viewer.

### Abundance estimation and mapping
The relative mean coverage of the scaffolds calculated from the mapping data described above were used as estimators of the scaffold abundance in the sample. The taxonomy of all non-viral scaffolds was retrieved using the same Lowest Common Ancestor methodology than previously published for the same dataset[14]. For co-occurrence analysis, the abundance of pairs of eukaryotes and viruses present in at least two samples were compared by spearman correlations. The resulting $p$ values were corrected using Benjamini & Hochberg FDR correction.

For in-between sample comparisons of viruses, read longer than 30 nucleotides were mapped to viral sequences from all samples with Bowtie 2 and a minimum quality filter of 30 was applied with SAMtools. Then, only scaffold with more than 10 kb covered was considered.

### Phylogenetic analysis
For the selected marker genes, individual gene trees were built from reference genomes only. Multiple alignments were performed using MAFFT (v7.407)[85], removal of divergent regions with ClipKIT[86] and models estimations[87] and tree inference using IQ-TREE (v1.6.12)[88] (options "-bb 1000"[89], "-bi 100" and "-m MFP"). The best model was VT + F + R4 for the TFIIS tree, LG + F + G4 for the MCP and LG + F + R5 for all the other marker genes. A global tree was calculated by a partitioned analysis[90] to include genomes with missing data. In addition, individual gene trees were computed with the same options and models.

To identify the marker genes in the permafrost data, PSI-BLAST was used to align reference marker genes to the viral ORFs (initial $E$ value ≤10[−5]). Next, in order to reduce the number of paralogs of the marker genes, we defined a second stringent $E$ value threshold the following way: $E$ values of all second matches for scaffolds with multiple copies were sorted in ascending order, then the stringent threshold was defined based on the first quartile (Supplementary Table 3). Finally, only the best match per scaffold was kept for phylogenetic reconstruction if it was better than the stringent threshold for this gene.

The 7 marker genes were aligned using PASTA[91], clipped with ClipKIT and concatenated by Catsequences[92]. The global tree was then inferred by IQ-TREE with ultrafast bootstraps options "-bb 1000" "-bi 200" and "-spp -m MFP" that calculates the best model per marker gene. Tree visualization was handled using FigTree (http://tree.bio.ed. ac.uk/software/FigTree/) and the Itol web server[93].

### Terrestrial *Nucleocytoviricota* distribution
We downloaded 1835 terrestrial assemblies from the JGI IMG/M[41] database (Supplementary Data 6) from March 2021, of which we kept only contigs ≥10 kb reducing the analysis to 1502 datasets. The ORFs were predicted using GeneMark using the metagenomic model as previously. *Nucleocytoviricota* sequences were extracted as described above (see Retrieving viral sequences). The same method than previously described (see Phylogenetic analysis) was applied to search for marker genes for phylogeny. Reference and metagenomic marker genes were aligned using MAFFT with the "−auto" option. Amsacta moorei entomopoxvirus, Variola virus and Cyprinid herpesvirus 2 were included in the analysis. The alignments were clipped with ClipKIT and concatenated for a partitioned analysis. Empirical models for each partition were inferred by Modelestimator[94]. Finally, the trees were computed using IQ-TREE (with options -bb 1000 -bi 200).

### Phylogenetic analyses of translation-related genes
A dataset of proteins was built using a combination of *Nucleocytoviricota* ORFs, corresponding BLAST matched proteins from the nr database and reference proteins from specific databases. The latter includes UniProt reviewed proteins of domains IPR001412 (class I aminoacyl-tRNA synthetases) and IPR006195 (class II aminoacyl-tRNA synthetases). The multiple alignments were performed using PASTA[91] or MAFFT[85] and trimmed with ClipKit[86]. The tree was then computed by IQ-TREE[88] with options -bb 5000 -bi 200 -m TEST.

### Reporting summary
Further information on research design is available in the Nature Research Reporting Summary linked to this article.

## Data availability
Large genome fragments and annotations were deposited to the EBI under the study PRJEB47746 with the following accessions: ERS10539964, ERS10539963, ERS10539962, ERS10539961, ERS10539960, ERS10539959, ERS10539958, ERS10539957. Accession codes of complete datasets can be found in Supplementary Table 1. In addition, previously published public data used for analysis includes: Genbank NR (from June 2020), GVMAGs (https://figshare.com/s/14788165283d65466732 and https://genome.jgi.doe.gov/portal/GVMAGs/GVMAGs.home.html), VOG orthogroups (https://vogdb.org/), Refseq protein database (from March 2020), EggNOG (v5), GEVEs (https://zenodo.org/record/3975964#.XzFj0hl7mfZ), JGI IMG/M (database 2021, https://img.jgi.doe.gov/). Source data are provided with this paper.

## Code availability
Custom scripts codes[95] used in this study can be accessed here: https://doi.org/10.6084/m9.figshare.20101850.

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

## Acknowledgements

We thank the PACA Bioinfo platform for computing support. Regarding the samples from Rigou et al.[14] used in this study, we would like to thank Alexander Morawitz for collecting the Kamchatka soil samples and Eugène Christo-Foroux for processing the sample and performing DNA

extraction, Dr. Jens Strauss and Dr. Guido Grosse for providing the Yukechi permafrost samples and Dr. Karine Labadie for supervising the sequencing on the Genoscope platform. We also thank François Enault and Hugo Bisio for carefully reading the manuscript. This work was supported by the CNRS Projets de Recherche Conjoints (PRC) grant (PRC1484-2018) to C.A. S.R. is supported by a doctoral fellowship obtained from Aix-Marseille University.

## Author contributions

Conceptualization: M.L., C.A., and J-M.C.; Methodology: S.R. and M.L.; Software: S.R.; Validation: S.R., M.L., and S.S.; Formal analysis: S.R.; Resources: J-M.C. and S.S.; Data curation: S.R.; Writing – original draft: S.R. and M.L.; Writing – review and editing: S.R., C.A., J-M.C., and M.L.; Visualization: S.R. and M.L.; Supervision: M.L.; Project administration: M.L.; Funding acquisition: C.A. and J-M.C.

## Competing interests

The authors declare no competing interests.
