## [Peer Review File · Nature Communications]

Past and present giant viruses diversity explored through permafrost metagenomicsReviewers' Comments:

Reviewer #1:

Remarks to the Author:

The manuscript by Rigou et al describes the diversity and genomic potential of Nucleocytoviricota members in a number of permafrost metagenomic sequence datasets. The samples show high heterogeneity in Nucleocytoviricota abundance, with as much as 12% of the total community in one sample, while only 2 sequences originated from giant viruses in another. Authors were also able to assemble a number of complete, circularized genomes from these datasets.

The methods, analysis and interpretation are sound in my opinion. I appreciate the thorough phylogenetic and functional analysis - although the manuscript seems excessively long. For example, I am not sure if the 'worldwide nucleocytoviricota distribution' section adds any key insight to the manuscript.

I have a few minor comments:

Line 101: produced by....??

Line 129-30 :what cutoffs and parameters were used for viralrecall?

In Figure 2, 'cellular, orfans, viral' don't align properly with the markings/best hits indicated in the figure.

Line 317: what do 'outliers' mean here?

Figure 5: how was the 'placement' of the marker genes conducted? What method, and what was the reference tree? Details needed.

Line 676: Psiblast should be PSI-BLAST.

Reviewer #2:

Remarks to the Author:

The paper by Rigou and colleagues is an interesting examination of the giant viruses found in permafrost. I have alot of little comments but my overall feeling is that this is a very good study and that this will ultimately be an appropriate journal for this work to appear in.

Comments:

The language is a little rough in a few areas - just small things that need to be cleaned up.

For example

Line 11. lesser would be better as less

line 12. does "two of them" refer to the soils or to viruses?

line 15. 12% of the total community? which community. And more correctly do you mean 12 % of the total sequence space?

line 18. you can delete "quite"

line 22. you can delete "also"

line 24 - 26. I am not 100% clear on what this last sentence means and I feel like there could be a more clear ending to the abstract - sorry, I have no suggestions at this time

Line 29. Remove "The" at the start.

Going forward I will reduce the grammatical edits but the comments above are examples of little things found through the entire document.

Line 33. I understand the use of "epigenetic" here but to microbiologists it means something else. Is there a better way to say this?

Line 36. I think there is a word missing in here. should "of" come after hundreds?

Lines 52/54. It seems like you are telling contradictory stories about the Phycodnaviridae here. Is this just a confusion of wording?

Line 61. Maybe I am the only one but I am tired of "viral dark matter". We know what viruses are doing (generally) and that they are doing it. So they are no longer "dark matter" can we just change this to viruses?

Line 71. The reference and the way it is present here made me think there is only one paper addresses giant viruses in metatranscriptomes, but that is not the case.

A very quick list could include Moniruzzaman et al (DOI: 10.1038/ncomms16054); Needham et al. 2019 (DOI: 10.1098/rstb.2019.0086); Gann et al (DOI: 10.3389/fmicb.2021.664189).

Onto the next line there have also been giant viruses recovered from metatranscriptomes in Sphagnum bogs (Stough et al 2081, DOI: 10.1128/AEM.01124-18) and large lakes (Pound et al. 2020, DOI: 10.3389/fmicb.2020.00338). That the majority of these studies ignore Pitho and Pandora is perhaps a major factor in making your work here so important. I think (gently) highlighting the oversight of others here really builds your case.

Line 80. Can this not also hamper assembly in marine systems?

Line 81 - 89. I know this is not the tenet of your study, but it makes me wonder why these are grouped so closely by phylogeneticists with the other giant viruses.

Line 94. I would change "is a great source" to "has"

Line 96. 12% of the total coverage is different than 12% of the total community (line 15)...and what percentage of the total raw reads is this?

Line 101. The reference here is a BioRxiv paper - can we not just add the text needed?

Line 108 (Figure S1). The figure legends for the supplemental materials need to provide more details. Just like within the main paper I think they should stand alone.

Line 121. In the figure legend the threshold is apparently set to 35%. This seems low to me or I am not understanding it. The three senior authors have a much better understanding of this than I ever could so I am guessing it is ok but I would like to come away from this understanding it.

Line 157. Are these really "genomes" or large fragments of genomes.

Figure 2. I spent some time sorting out where the names come from in here so perhaps that can be clarified in the figure legend.

Also, there are some misalignments in the labels (insets of genomes....looks like they should all be

Virus/cellular/ORFans but Krossosvirus, Marosvirus, and down have been shifted).

Line 182 - 183. I think this is interesting. Other giants had this problem when they were first described (e.g., AaV). That they no regularly show up as abundant in metaGen/Transc studies highlights the importance of studying them. Is it possible Staryvirus is also globally distributed? oops. that might be yet to come.

Line 226. Is free the right word here? Do you mean free outside a cell???

Figure 4. Firstly I am having a really hard time reading this (working off a print copy). That said it seems that alot of diversity in the giant virus realm is being missed here - I am curious as to why? For example, PbCV1 is not on the tree that I can see. Given it is the best / longest studied of the giants and a fresh water representative I thought I would find it. Should there not be freshwater versions on here? Likewise Aureococcus virus, which shows up as abundant in marine systems and is well studied is also not on here. Perhaps these cause the tree to break in an unfortunate way, but it leaves me to wonder,

In parallel the organic lake metagenomic assemblies are on here and presented as intact, isolated viruses when they are in fact metagenomic assemblies. I get that many people do this but that does not make it correct. Leaving them on is fine, but honesty in what they are (and maybe that is hidden in the methods and I missed it) is needed. I think the same goes for Kloesneovirus - assembly, not isolate (again, if I missed something in the literature my apologies).

Line 290. Ah. Excellent!

Figure 5. I find several of the shades of grey (maybe light purple) look very similar. Can this be tweaked?

Figure 8. I wish the legend told me what the take away was from the figure.

Line 543-545. This thought needs a reference.

Line 574. OK, a large pet peeve of mine. How were the samples collected, extracted and sequenced. I appreciate that the bioRxiv paper you cite earlier might have that information, but that is an unreviewed paper at this point. I think this needs to be here. All the pages of pipelines and version controls and scripts mean nothing if the samples are not collected and processed properly. This needs to be well documented with sequencing depth, any preprocessing. etc. For example, contamination control (line 641) might be an informatic step, but it is also very much a sample collection and processing step.

Overall this is a very good paper and I look forward to seeing the final published version

Reviewer #3:

Remarks to the Author:

The authors analyzed permafrost metagenome data, from which they assembled unique viral genomes, found a unique sample with high proportion of Nucleocytoviricota, and analyzed genes found in their MAGs. Permafrost metagenome data may be important, and each result seems to be interesting, but they are not closely related each other, which makes the manuscript descriptive and conclusion blurred. At least, authors should remove latter part, in which functional and phylogenetic analysis was performed. This part is general analysis of nucleocytoviricota and the permafrost data has little contribution for the conclusion. Instead, the authors have to improve early part of analysis, which have questionable points. My major concerns are below.

L100 The authors discussed contamination level of their pipeline based on CheckM results with Fig S1. However, Fig S1 showed that bins have contamination, which are reduced in scaffolds over 50 kbp. Therefore, Fig S1 does not provide any information about reliability of the author's pipeline. In addition, the Fig S1A indicates that bins have substantial contamination, which means that binned reads used for the assembly also have contamination. General assemblers take read population as reference for assembly. If contaminated reads are provided as pure source, the assembler may produce artificial assembly. The authors have to examine the possibility more carefully.

L116 This section explained the extracting process of nucleocytoviricota genome. The authors only consider the number of matches, but the length of each scaffold should be considered. In addition, the threshold of viral genome is arbitrary. Statistical evaluation should be used.

L154 As I indicated above, the authors have to carefully consider the possibility of artificial assembly. Otherwise, following analysis is useless. Also, the authors must not name incomplete MAGs, which will disturb following studies. Even if the authors obtained nearly complete genome, MAGs should be named according to their relatives, rather than completely new and irrelevant name. For example, the largest MAGs in the manuscript is distantly but clearly related to pithovirus or orpheovirus, so that it should be named based on these two virus name.

L192 It is interesting that one sample have high proportion of nucleocytoviricota. However, large difference between samples implies sampling bias. Unlike water samples, where viral particles freely move around, soil samples may have biased distribution of viral particles in samples. The authors have to carefully examine the potential cause of this high virus proportion. At least, the authors should investigate taxonomic composition of all domains of life and viruses.

L572 In Material and Methods, the authors described how they process the data, but quantitative data is often lacked in the description. One example is in L656, it is unclear what "potential coverage drop" means.

Minor concerns are below

Throughout the manuscripts, the authors have to write software's name with proper manner. For example, "CheckM" is the correct name and not "checkm".

Throughout the manuscripts, the authors use the expression like "described in (X)". However, it should be "described in a previous study (X)".

Reviewer #1 (Remarks to the Author):

The manuscript by Rigou et al describes the diversity and genomic potential of Nucleocytoviricota members in a number of permafrost metagenomic sequence datasets. The samples show high heterogeneity in Nucleocytoviricota abundance, with as much as 12% of the total community in one sample, while only 2 sequences originated from giant viruses in another. Authors were also able to assemble a number of complete, circularized genomes from these datasets.

R1.1

The methods, analysis and interpretation are sound in my opinion. I appreciate the thorough phylogenetic and functional analysis - although the manuscript seems excessively long. For example, I am not sure if the 'worldwide nucleocytoviricota distribution' section adds any key insight to the manuscript.

We thank the reviewer #1 for this positive evaluation of our work. As requested we have reorganized the results section and removed some parts (including the paragraph on the MCP network) and drastically reduced others such as the "worldwide nucleocytoviricota distribution" section. We now only make comparison to other terrestrial datasets and focus on the primary point that permafrost samples are richer in giant viruses than other soils samples. In the end the manuscript is reduced by roughly 15%.

I have a few minor comments:

R1.2

Line 101: produced by....??

The referred manuscript by Rigou et al initially deposited in BioRxiv that describes the samples is now formally accepted and in press in the MicroLife journal. The new reference is the following:

Sofia Rigou, Eugène Christo-Foroux, Sébastien Santini, Artemiy Goncharov, Jens Strauss, Guido Grosse, Alexander N. Fedorov, Karine Labadie, Chantal Abergel, Jean-Michel Claverie. Metagenomic survey of the microbiome of ancient Siberian permafrost and modern Kamchatkan cryosols; DOI: 10.1093/femsml/uqac003

As the manuscript is still in press we are sharing with you the MicroLife editor acceptance letter as well as the final accepted manuscript.

R1.3

Line 129-30 :what cutoffs and parameters were used for viralrecall?

We used the default ViralRecall minimal score threshold of 0. This default value seems indeed reasonable as it well discriminates viral and cellular contigs based on our control database (see below):

With that default threshold, 94% of viral contigs detected by our method are also confirmed by ViralRecall. We now add the following sentence in the Materials and Methods section:

“For comparison, we also tested the ViralRecall tool (35) that confirmed 1848 out of the 1973 (94%) scaffolds detected by our pipeline. “

R1.4

In Figure 2, ‘cellular, orfans, viral’ don’t align properly with the markings/best hits indicated in the figure.

Sorry for this alignment mistake in the figure. This is now corrected.

R1.5

Line 317: what do ‘outliers’ mean here?

This section is no longer part of the manuscript.

R1.6

Figure 5: how was the ‘placement’ of the marker genes conducted? What method, and what was the reference tree? Details needed.

The details on the procedure are in the “Phylogenetic analysis” paragraphs in the Materials and Methods section including the evolutionary models used for each marker gene, the different options used in IQ-TREE for phylogenetic reconstructions and the fact that we performed a partitioned analysis. We also added the following information in the figure legend (now Fig. 3):

“Consensus of 1000 bootstrapped trees calculated by IQ-TREE through a partitioned analysis on 7 marker genes. The models used were the following: LG+F+R5 for the packaging ATPase and the MCP, LG+F+R6 for the RNA polymerase subunits RPB1 and RPB2, LG+F+R7 for the primase D5, LG+R+R10 for the DNA polymerase B and VT+F+R7 for VLTF3.”

R1.7

Line 676: Psiblast should be PSI-BLAST.

This is now corrected.

Reviewer #2 (Remarks to the Author):

The paper by Rigou and colleagues is an interesting examination of the giant viruses found in permafrost. I have a lot of little comments but my overall feeling is that this is a very good study and that this will ultimately be an appropriate journal for this work to appear in.

We are very grateful to reviewer #2 for this very positive comment.

Comments:

R2.1

The language is a little rough in a few areas - just small things that need to be cleaned up.

For example

Line 11. lesser would be better as less

Corrected

line 12. does "two of them" refer to the soils or to viruses?

This refers to viruses. We slightly changed the sentence to avoid confusion:

"Less is known about giant viruses from soil even though two of them, belonging to two different viral families, were reactivated from 30,000-y-old permafrost samples."

line 15. 12% of the total community? which community. And more correctly do you mean 12 % of the total sequence space?

Thanks for that remark. We agree that this needs to be more precise. We changed the abstract with the following sentence:

"Through permafrost metagenomics we reveal a unique diversity pattern and a high heterogeneity in the abundance of giant viruses, representing up to 12% of the sum of sequence coverage in one sample."

We also added this sentence in the main text:

"Nucleocytoviricota scaffolds corresponded to 12% of the R sample sequence coverage (Fig. 2) and 17% of total reads mapped on scaffolds over 10 kb (4% of all raw reads)."

line 18. you can delete "quite"

Corrected

line 22. you can delete "also"

Done

R2.2

line 24 - 26. I am not 100% clear on what this last sentence means and I feel like there could be a more clear ending to the abstract - sorry, I have no suggestions at this time

The functional annotation of permafrost nucleocytoviricota shows a patchwork of predicted functions and a large proportion of genes with unknown functions. We believe the sentence is clear enough. The abstract ends following way:

“The annotation of the permafrost viral sequences revealed a patchwork of predicted functions amidst a larger reservoir of genes of unknown functions. Finally, the phylogenetic reconstructions not only revealed gene transfers between cells and viruses, but also between viruses from different families. “

Line 29. Remove "The" at the start.

Done

Going forward I will reduce the grammatical edits but the comments above are examples of little things found through the entire document.

We have rewritten some parts of the manuscript hoping to solve these grammatical problems.

R2.3

Line 33. I understand the use of "epigenetic" here but to microbiologists it means something else. Is there a better way to say this?

This is the appropriate technical term but we have changed the definition of “epigenetic” for more clarity:

“The microbial community of the surface cryosol is in some cases subject to freezing and thawing of the soil every year (2) whereas communities from deeper layers are trapped as the sediments are deposited (syngenetic permafrost) or as the sediment freezes (epigenetic permafrost).”

R2.4

Line 36. I think there is a word missing in here. should "of" come after hundreds?

Corrected

R2.5

Lines 52/54. It seems like you are telling contradictory stories about the Phycodnaviridae here. Is this just a confusion of wording?

We agree. We actually decided to remove the sentence as there are very few Phycodnaviridae scaffolds recovered from the cited study in glacial environments.

R2.6

Line 61. Maybe I am the only one but I am tired of "viral dark matter". We know what viruses are doing (generally) and that they are doing it. So they are no longer "dark matter" can we just change this to viruses?

We changed to “viruses” instead of “viral dark matter” as requested.

R2.7

Line 71. The reference and the way it is present here made me think there is only one paper addresses giant viruses in metatranscriptomes, but that is not the case.

A very quick list could include Moniruzzaman et al (DOI: [10.1038/ncomms16054](https://doi.org/10.1038/ncomms16054)) SMASH; Needham et al. 2019 (DOI: [10.1098/rstb.2019.0086](https://doi.org/10.1098/rstb.2019.0086)) SMASH; Gann et al (DOI: [10.3389/fmicb.2021.664189](https://doi.org/10.3389/fmicb.2021.664189)) SMASH.

Onto the next line there have also been giant viruses recovered from metatranscriptomes in Sphagnum bogs (Stough et al 2081, DOI: [10.1128/AEM.01124-18](https://doi.org/10.1128/AEM.01124-18)) SMASH and large lakes (Pound et al. 2020, DOI: [10.3389/fmicb.2020.00338](https://doi.org/10.3389/fmicb.2020.00338)) SMASH. That the majority of these studies ignore Pitho and Pandora is perhaps a major factor in making your work here so important. I think (gently) highlighting the oversight of others here really builds your case.

We thank the reviewer for pointing out these references, some of which had escaped our attention. Some are indeed perfectly appropriate and relevant to our argument. We now cite the following references: Moniruzzaman et al 2017 (<http://dx.doi.org/10.1038/ncomms16054>), Gann et al 2021 (<https://doi.org/10.3389/fmicb.2021.664189>) and Pound et al (<https://doi.org/10.3389/fmicb.2020.00338>).

R2.8

Line 80. Can this not also hamper assembly in marine systems?

Microbiome complexity can indeed hamper the assembly of marine systems metagenomes although to a lesser extent than soil biomes that are usually more diverse. For instance in the Figure 2 of the Rodriguez et al work (Nonpareil 3: Fast Estimation of Metagenomic Coverage and Sequence Diversity, <https://doi.org/10.1128/mSystems.00039-18>) one can see that most of the soil samples are more diverse than marine ones. We also contacted the EBI helpdesk in charge of the Mgnify database that confirmed that there are still very few soil samples in the peptide database, derived from assembled datasets. They explained that this was mainly due to the fact that de novo assembly on soil sample data was very difficult because of the enormous species diversity contained within the samples.

R2.9

Line 81 - 89. I know this is not the tenet of your study, but it makes me wonder why these are grouped so closely by phylogeneticists with the other giant viruses.

We absolutely agree with the reviewer on this point. In fact, we actually discuss it in the penultimate paragraph of the discussion section. But as pointed out by the reviewer this is not the main point of this work, so we preferred to focus on the diversity of giant viruses in permafrost rather than on taxonomic debates.

R2.10

Line 94. I would change "is a great source" to "has"

Corrected

R2.11

Line 96. 12% of the total coverage is different than 12% of the total community (line 15)...and what percentage of the total raw reads is this?

We agree that this had to be clarified. The percentage of total raw reads is 4% which correspond to 17% of all the reads mapping on contigs over 10 kb (the ones used in this study). This information is now added in the manuscript:

“Nucleocytoviricota scaffolds corresponded to 12% of the R sample sequence coverage (Fig. 2) and 17% of total reads mapped on scaffolds over 10 kb (4% of all raw reads).”

R2.12

Line 101. The reference here is a BioRxiv paper - can we not just add the text needed?

As answered to reviewer #1 the publication is now accepted and in press in the MicroLife journal (see point R1.2). We provide you with the acceptance letter and the accepted version of the manuscript.

R2.13

Line 108 (Figure S1). The figure legends for the supplemental materials need to provide more details. Just like within the main paper I think they should stand alone.

The Figure S1 has been replaced to address some concerns raised by other reviewers (see R3.1). We believe that the data are now solid and clearer, and that the accompanying legend is more detailed. We have also modified some of the legends for the other figures (main text and supplements) to provide more details.

R2.14

Line 121. In the figure legend the threshold is apparently set to 35%. This seems low to me or I am not understanding it. The three senior authors have a much better understanding of this than I ever could so I am guessing it is ok but I would like to come away from this understanding it.

Thanks for this remark. The threshold was indeed set to 35% sequence identity to ensure that we did not hit the “twilight zone”. But we also applied a more statistically sound E-value threshold ($< 10^{-5}$) that we did not mention in the first version of the manuscript. We apologize for this oversight. This is now duly mentioned in the manuscript (Fig. 1 legend and Materials and methods).

R2.15

Line 157. Are these really “genomes” or large fragments of genomes.

These are indeed large genomes fragments except for the largest one of 1.6 Mb that is probably complete since we were able to circularize it. We thus changed the subtitle to “Large viral genome fragments”.

R2.16

Figure 2. I spent some time sorting out where the names come from in here so perhaps that can be clarified in the figure legend.

The names actually all come from Greek related to “vases” as for “Pithos”. However, since the other reviewers also complained about naming genomes fragments (see R3.3) we changed the naming of the large genome fragments with the phylogenetic group they belong to, the sample type and the scaffold code (for instance: Pithovirus/Orpheovirus Permafrost:N_b1782_k1). We kept the name “Pithovirus/Orpheovirus Permafrost:Hydrivirus” (see new Fig. 4) though as this correspond to a complete genome. Previous metagenomic studies have used names even for not isolated viruses (see for instance “Klosneuviruses”). If required we can also change Hydrivirus to a scaffold code.

Also, there are some misalignments in the labels (insets of genomes....looks like they should all be Virus/cellular/ORFans but Krossosvirus, Marosvirus, and down have been shifted).

This is now corrected (see R1.4).

R2.17

Line 182 - 183. I think this is interesting. Other giants had this problem when they were first described (e.g., AaV). That they no regularly show up as abundant in metaGen/Transc studies highlights the importance of studying them. Is it possible Staryvirus is also globally distributed? ooops. that might be yet to come.

This is an interesting question for which we do not have a definitive answer yet. The former “Staryvirus” now named “Unknown Permafrost:M_b2437_k1” is absent from the other terrestrial datasets we have examined in the JGI IMG/G database. For this reason, we do not elaborate on this point in the manuscript. However, not finding this virus in terrestrial metagenomics datasets does not give a definitive answer on whether it is globally distributed or not. For instance the Pandoraviridae that were isolated from various locations and environments are still not significantly found in metagenomes.

R2.18

Line 226. Is free the right word here? Do you mean free outside a cell???

We changed the sentence to:

“Altogether, this suggests that most of the discovered permafrost Nucleocytoviricota scaffolds correspond to bona fide unintegrated viruses.”

R2.19

Figure 4. Firstly I am having a really hard time reading this (working off a print copy). That said it seems that alot of diversity in the giant virus realm is being missed here - I am curious as to why? For example, PbCV1 is not on the tree that I can see. Given it is the best / longest studied of the giants and a fresh water representative I thought I would find it. Should there not be freshwater versions on here? Likewise Aureococcus virus, which shows up as abundant in marine systems and is well studied is also not on here. Perhaps these cause the tree to break in an unfortunate way, but it leaves me to wonder,

PbCV1 is actually present in the phylogenetic tree (in now Fig. 3) next to Acanthocystis turfacea Chlorella virus 1. Given the “composite” nature of AAV with matches in many different families (see R2.17) we decided not to include it in the phylogenetic as we thought it would make the signal noisier and would lower bootstrap values.

R2.20

In parallel the organic lake metagenomic assemblies are on here and presented as intact, isolated viruses when they are in fact metagenomic assemblies. I get that many people do this but that does not make it correct. Leaving them on is fine, but honesty in what they are (and maybe that is hidden in the methods and I missed it) is needed. I think the same goes for Kloesneovirus - assembly, not isolate (again, if I missed something in the literature my apologies).

We agree with this comment. We thus now mark viruses coming from metagenomic assemblies in the Fig. 3 and added the following sentence to the legend:

“One should note that reference genomes coming from bins of previous metagenomic studies (marked with a black dot) are less reliable than the genomes of isolated viruses.”

R2.21

Line 290. Ah. Excellent!

Thanks for this enthusiasm. However as the two other reviewers thought this part of the manuscript was too long we reduced it to the minimum to emphasize only the main points (see R1.1).

R2.22

Figure 5. I find several of the shades of grey (maybe light purple) look very similar. Can this be tweaked?

We are unsure of what shades the reviewer is referring to. In any case the former Figure 5A is no longer part of this manuscript and the Figure 5B is now in supplements (new Fig S11A).

Figure 8. I wish the legend told me what the take away was from the figure.

This figure is no longer part of the manuscript.

R2.23

Line 543-545. This thought needs a reference.

We now add the following reference from Munson-McGee et al: A virus or more in (nearly) every cell: ubiquitous networks of virus-host interactions in extreme environments. DOI: 10.1038/s41396-018-0071-7

R2.24

Line 574. OK, a large pet peeve of mine. How were the samples collected, extracted and sequenced. I appreciate that the bioRxiv paper you cite earlier might have that information, but that is an unreviewed paper at this point. I think this needs to be here. All the pages of pipelines and version controls and scripts mean nothing if the samples are not collected and processed properly. This needs to be well documented with sequencing depth, any preprocessing. etc.

For example, contamination control (line 641) might be an informatic step, but it is also very much a sample collection and processing step.

Absolutely. The referred paper is now formally accepted in MicroLife (see R1.2 and R2.12) and the “Sample collection and preparation” paragraph in the Materials and Methods section has been extended to better explain the procedure. We provide you now with the acceptance letter as well as the final version of the manuscript.

R2.25

Overall this is a very good paper and I look forward to seeing the final published version

We again thank the reviewer for his positive and constructive comments on this work.

Reviewer #3 (Remarks to the Author):

The authors analyzed permafrost metagenome data, from which they assembled unique viral genomes, found a unique sample with high proportion of Nucleocytoviricota, and analyzed genes found in their MAGs. Permafrost metagenome data may be important, and each result seems to be interesting, but they are not closely related each other, which makes the manuscript descriptive and conclusion blurred. At least, authors should remove latter part, in which functional and phylogenetic analysis was performed.

This part is general analysis of nucleocytoviricota and the permafrost data has little contribution for the conclusion. Instead, the authors have to improve early part of analysis, which have questionable points. My major concerns are below.

To follow the reviewer advice we drastically reduced the functional and phylogenetic part of the manuscript. We mostly centered these analyses to emphasize the contribution of the Nucleocytoviricota discovered in the permafrost compared to already known Nucleocytoviricota from other environments. We also have improved the early part of the study as requested that will hopefully dissipate the reviewers main concerns (see next point).

R3.1

L100 The authors discussed contamination level of their pipeline based on CheckM results with Fig S1. However, Fig S1 showed that bins have contamination, which are reduced in scaffolds over 50 kbp. Therefore, Fig S1 does not provide any information about reliability of the author's pipeline. In addition, the Fig S1A indicates that bins have substantial contamination, which means that binned reads used for the assembly also have contamination. General assemblers take read population as reference for assembly. If contaminated reads are provided as pure source, the assembler may produce artificial assembly. The authors have to examine the possibility more carefully.

Contigs' binning is now a standard procedure in metagenomic studies and nearly all environmental giant virus studies rely purely on bins but we totally agree with the reviewer that contamination biases have to be carefully examined. We thus reinforced contamination controls in our assembly procedure and provide a new supplementary figure (Fig. S1) that we believe prove that we worked with a clean dataset. We started with a standard assembly with a very low contamination rate according to CheckM (0.004% potential chimeras, Fig S1A), whatever the contig size. We then performed binning and noticed a higher contamination level (gray bars in Fig. S1). We thus performed a second de novo assembly on individual sub-datasets corresponding to the bins. To avoid chimeras we used SPAdes that consider the coverage of the contigs and also the "-meta" options to resolve potential conflicts when necessary. This resulted in a very clean second assembly as shown by the low contamination rate detected by CheckM (on average 0.005% potential chimeras and none at the strain level, blue bars in Fig S1A).

Next, to reinforce our confidence that we did not produce artificial assemblies we performed a second control not using CheckM. This was done using previously published complex and strain heterogeneous mock communities. We performed the different assembly steps previously described and aligned the resulting assembled sequences on the known reference genomes from the mock community (Fig S1B). This revealed again a negligible contamination level of 0.2% on the final assembly (Fig. S1B, blue bars).

This method is similar to a previous work done by others (Lui LM et al. A method for achieving complete microbial genomes and improving bins from metagenomics data. PLOS Computational Biology 17:e1008972.), without the circularization step, resulting in high quality assembled metagenomes.

We added the following paragraphs in the main text to describe the different controls:

"We first performed an assembly of the reads (Table S2) resulting in clean contigs with very few potential chimeras (0.004%) and no strain level chimera as estimated by CheckM (32) (Fig. S1A). CheckM applies here as our complete dataset is mainly (90%) prokaryotic (14). Next, as is custom in metagenomics studies, we performed a binning of the contigs to obtain less fragmented assemblies (33). This revealed potential

chimeras (Fig. S1A). We thus chose not to consider bins as unique organisms but instead, we used binning as a procedure to decrease complexity in our datasets. More precisely, the reads were first separated according to the bin they belonged to. Next, a second de novo assembly was made within each bin. This resulted in significantly longer scaffolds and a larger total assembly (Table S2) while keeping contamination at a negligible level (on average 0.005% potential chimeras and again none at the strain level, Fig. S1A). Thus, our method significantly gained in reliability by lowering the proportion of chimeras in comparison to conventional binning, while providing longer assembled sequences compared to standard assemblies. A similar strategy recently produced high quality bacterial and megaphage genomes (34).

To further validate this strategy, we applied the same assembly method on three complex mock communities generated by a previous study (35). Aligning the reference genomes used in that study on the resulting assembled sequences revealed a similar pattern: a clean first assembly, a noisier binned assembly and a clean final assembly (Fig. S1B). The proportion of chimeras in the final scaffolds accounts for only 0.2%.”

R3.2

L116 This section explained the extracting process of nucleocytoviricota genome. The authors only consider the number of matches, but the length of each scaffold should be considered. In addition, the threshold of viral genome is arbitrary. Statistical evaluation should be used.

We initially used the proportion of matches to take into account scaffold length but realized that counts instead gave better classification performances with less false positives and false negatives. This information is now available in Fig. S3. However, we agree that scaffold length might indeed be an influencing factor. To consider that issue we analyzed the false/true positives/negatives statistics in different contig size ranges (Fig. S2B). From this, one can see that the smaller the contig the higher the proportion of false negatives. However the trend is much less clear with false positives. Thus we can conclude that we may miss true Nucleocytoviricota from our permafrost datasets using the counts but that we are most likely not polluting the data with false Nucleocytoviricota sequences.

R3.3

L154 As I indicated above, the authors have to carefully consider the possibility of artificial assembly. Otherwise, following analysis is useless. Also, the authors must not name incomplete MAGs, which will disturb following studies. Even if the authors obtained nearly complete genome, MAGs should be named according to their relatives, rather than completely new and irrelevant name. For example, the largest MAGs in the manuscript is distantly but clearly related to pithovirus or orpheovirus, so that it should be named based on these two virus name.

In addition to the supplementary control we performed to estimate potential artificial assemblies (see R3.1), we manually scrutinized sequence coverage for potential misassemblies of the large MAGs (see Materials & Methods section “Large genomes assembly verification and circularization”). All MAGs presented in the new Fig. 4 were validated.

According to the reviewer suggestion we renamed the incomplete MAGs using the phylogenetic group they belong to, the sample type and the scaffold code (see R2.16). Regarding the largest one, we kept the Pithovirus/Orpheovirus relatedness in the name: Pithovirus/Orpheovirus Permafrost: Hydrivirus (see new Fig. 4). “Hydri” comes from the Greek vase “hydria” to remind of “Pithos”.

R3.4

L192 It is interesting that one sample have high proportion of nucleocytoviricota. However, large difference between samples implies sampling bias. Unlike water samples, where viral particles freely move around, soil samples may have biased distribution of viral particles in samples. The authors have to carefully examine the potential cause of this high virus proportion. At least, the authors should investigate taxonomic composition of all domains of life and viruses.

The investigation of the taxonomic distribution of all domains in these samples has been performed in a previous study (Rigou et al, now accepted in *Microlife*, see R1.2). The taxonomy of the contigs determined through the Lowest Common Ancestor method at a 50% threshold of the DIAMOND BLASTP matches shows that the samples are indeed very heterogeneous in all aspects. It is only when looking at the functional profiling (clustering based on Pfam counts) that similarities in-between samples L, N and R from the same borehole can be observed.

In the present work, we showed a clear trend in correlation of the Eukaryota and Nucleocytoviricota abundance. Still, we did not get deeper into the analysis (for instance co-presence) because of lack of data. We can only point out the presence of Longamoebia in sample R and leave it as a probable influencer.

As noticed by the reviewer we cannot exclude a biased distribution of virus particles due to the intrinsic heterogeneous nature of soil samples as compared to water samples. The large amount of soil (20g) used to extract DNA might have tempered this risk though. We now discuss it in the manuscript:

“Nevertheless, we highlighted an important heterogeneity in Nucleocytoviricota proportions across the samples, in agreement with the already observed heterogeneity at various scales (domain, phylum, class, order and functional annotation) for all domains of life (14). This heterogeneity is probably the testimony of the absence of mixing between layers but also of a spatially heterogeneous microbiome. It has been pointed out that the bacterial community of agricultural soil changes at a centimeter-level (52). Thus, the heterogeneity we observe might translate a sampling bias although probably attenuated by the large amount of soil from each sample (20g) used for DNA extraction.”

R3.5

L572 In Material and Methods, the authors described how they process the data, but quantitative data is often lacked in the description. One example is in L656, it is unclear what “potential coverage drop” means.

We now provide more quantitative details on this aspect:

“Then we used the Integrative Genome Viewer (82) to scrutinize the positions where the coverage dropped under 3x (mainly due to ambiguous bases added during scaffolding).”

Minor concerns are below

R3.6

Throughout the manuscripts, the authors have to write software’s name with proper manner. For example, “CheckM” is the correct name and not “checkm”.

Thanks. This is now corrected.

R3.7

Throughout the manuscripts, the authors use the expression like “described in (X)”. However, it should be “described in a previous study (X)”.

Corrected.

Reviewers' Comments:

Reviewer #1:

Remarks to the Author:

The authors have adequately addressed all my concerns. I believe the modifications substantially improved the manuscript.

Reviewer #2:

Remarks to the Author:

The authors have done an excellent job in dealing with all my criticisms. I have had the opportunity to review these as well as the revised paper and am happy with the current version.

Thank you

Reviewer #3:

Remarks to the Author:

Previously, I proposed to the authors that they should focus on the topics tightly connected to the uniqueness of permafrost samples. However, I did not find any improvements for this point in the revised manuscript. Although the manuscript is little reorganized, the analysis itself is almost the same. Again, I agree that some results are potentially important, but still the analysis is fragmented, and conclusion is unclear. Particularly in figure 2, the authors must examine properties of the samples much deeper. For example, co-occurrence of NCLDV with eukaryotes is one way. Moreover, latter analysis such as phylogeny and functional enrichment have to be performed for each sample, and their difference should be discussed with timescale, geography, and other ecological parameters. The authors' reply to technical points I concerned before is also not convincing. Authors discussed contamination level of binning based on CheckM results. I agree CheckM partially help to know the contamination level of the MAG. However, the authors must be aware that the tool was developed for bacteria and archaea and no one proved the CheckM results can be equally applied to viruses. Furthermore, the authors described their alternative validation is similar to previously published one, named "Jorg". However, the method in the manuscript is fundamentally different from Jorg. Jorg aims to circularization of the genome, as circularity is one measurement of the bacterial genome completeness. Jorg takes only circularized genomes are considered as true assembly. In contrast, the authors skipped the circularization step, the crucial one for the Jorg. In addition, Jorg utilizes another method, mirabait and MIRA, for second mapping and assembly, respectively. Jorg's paper describes this mapping-based second step is more accurate than k-mer-based method, like SPAdes. The authors did not consider these points and only used SPAdes for first and second assemblies. Collectively, the authors statements did not provide sufficient evidence to support the quality of their assembly.

Below are additional major concerns that I did not point before.

Line 228: They insisted they revealed the diversity of the pithoviruses. However, they used contigs and not MAGs in their phylogenetic analysis. In the figure 4, contigs with one marker gene are included. In their method, split genomes cannot be rescued if first binning failed. Therefore, it is highly possible they originated from limited number of genomes, and the pithoviruses diversity was overrepresented.

Line 333: "patchwork of functions" is not surprising for NCLDVs. This part should be more carefully examined if there are statistically unique functions found in permafrost NCLDVs, using GO and KEGG.

Line 345: The authors did not provide reasonable information why they focus on histones. Thus, results are not related to the permafrost environment. This is also true for translation-related genes.

REVIEWER COMMENTS

The reviewers' comments are noted in black font, our responses are in red and citations to the manuscript in green.

Reviewer #1 (Remarks to the Author):

The authors have adequately addressed all my concerns. I believe the modifications substantially improved the manuscript.

We are pleased that the reviewer approved this version of the manuscript and would like to thank him/her for his/her valuable comments.

Reviewer #2 (Remarks to the Author):

The authors have done an excellent job in dealing with all my criticisms. I have had the opportunity to review these as well as the revised paper and am happy with the current version.
Thank you

We are very grateful to the reviewer for the comments that truly helped to improve our manuscript.

Reviewer #3 (Remarks to the Author):

Previously, I proposed to the authors that they should focus on the topics tightly connected to the uniqueness of permafrost samples. However, I did not find any improvements for this point in the revised manuscript. Although the manuscript is little reorganized, the analysis itself is almost the same. Again, I agree that some results are potentially important, but still the analysis is fragmented, and conclusion is unclear. Particularly in figure 2, the authors must examine properties of the samples much deeper. For example, co-occurrence of NCLDV with eukaryotes is one way.

The primary goal of this work was to highlight the diversity and abundance of Nucleocytoviricota in permafrost. The samples we explored in this work are thus not perfectly suited to reveal virus-host interactions with a high statistical power (limited number of samples, no replicates over time). That being said, we followed the reviewer advice and performed a co-occurrence analysis of permafrost Nucleocytoviricota and eukaryotic families, as shown in the Fig. S4B. We did find a handful of significant co-occurrences, one of which involves putative amoebic hosts, but also more unexpected associations between the divergent Pithoviridae revealed in this work and Hydrozoa as well as Cryptomonadaceae. This suggests a potentially wide range of hosts for Pithoviridae, as is now consensually admitted for Mimiviridae (see for instance <https://doi.org/10.3390/v10090506>). Although, one has to keep in mind that this type of correlative analyses might also reveal indirect interactions between viruses and eukaryotes. We added the following paragraph in the results section to present this new analysis:

“The relative proportion of giant viruses (Fig. 2) showed a strong correlation to the ones of Eukaryota. Precisely, Spearman correlation coefficients of $\rho=0.72$ for the sum of coverages (p -value=0.017, Fig. 2) and $\rho=0.83$ for the number of scaffolds (p -value=0.003) were observed. Such correlation could be explained by host-parasites dynamics. We therefore looked for potential co-occurrences of viral and eukaryotic families. Despite working with only 11 samples, we found significant associations (Fig. S4B), including two Pithoviridae-like viruses with Entamoebidae. More surprisingly, we also found two other Pithoviridae-like

associated with Hydrozoa. HGT between Mimiviridae and this eukaryotic class has already been observed (35). Finally, two other Pithoviridae-like were also found associated with Cryptomonadaceae. Although these eukaryotes are not known to be infected with giant viruses, metagenomics co-occurrence analyses showed association between cryptophytes and Mimiviridae (22) as well as viroplages (36).”

We also added the description of the methods we used for that purpose in the Materials and Methods section:

“The relative mean coverage of the scaffolds calculated from the mapping data described above were used as estimators of the scaffold abundance in the sample. The taxonomy of all non-viral scaffolds was retrieved using the same Lowest Common Ancestor methodology than previously published for the same dataset (14). For co-occurrence analysis, the abundance of pairs of eukaryotes and viruses present in at least two samples were compared by spearman correlations. The resulting p-values were corrected using Benjamini & Hochberg FDR correction.”

Finally, this new analysis suggested by the reviewer is now mentioned in the discussion:

“The co-occurrence analysis of Nucleocytoviricota and eukaryotes performed in this study linked Pithoviridae-like clades to amoeba. More surprisingly others were associated with Cryptomonadaceae and Hydrozoa. This widens the possible host range of Pithoviridae in the same way Mimiviridae infect various distant clades (51). However, co-occurrence might translate indirect correlation and not direct virus-host interactions.”

Moreover, latter analysis such as phylogeny and functional enrichment have to be performed for each sample, and their difference should be discussed with timescale, geography, and other ecological parameters.

Again, this study was designed to explore the diversity of Nucleocytoviricota in permafrost rather than to understand the ecological niches of these viruses. Nevertheless, in addition to the taxonomical diversity we observed in the viral content of the samples (Fig. 2), we now add a statistical analysis of functional enrichments between samples using Gene Ontology and Pfam domains as requested (Table S8). This ended up in virtually no function significantly enriched between samples after FDR correction when considering the same viral families. This indicates that genome content and ecological parameters are not directly correlated or, more likely, that the high proportion of genes with unknown functions and the limited number of samples prevent this from being revealed at this time.

As requested, we also analyzed the total functional content of the samples with all viral families mixed together in relation to ecological parameters. This is summarized in the new figure S12 where samples were clustered according to the Pfam domains content using a Bray-Curtis distance in relation with age, depth, soil type and nonpareil diversity (a proxy to community diversity). It is clear from this analysis that there is no obvious conclusion to draw on the relation between Nucleocytoviricota-encoded functions and ecological parameters. This is again most likely due to the number of samples studied which prevents revealing an overall trend.

We thus added the following in the results section:

“We also did not find specific functional enrichment when comparing samples to each other within the same viral families (Table S8). Likewise, when mixing all viral families together, ecological parameters do not discriminate samples based on Pfam annotations (Fig. S12). Altogether, this indicates that viral genome content and ecological parameters are not directly correlated or, more likely, that the high proportion of genes with unknown functions and the limited number of samples prevent this from being revealed at this time.”

and in the Materials and Methods section:

“The Pfam and GO term annotations were retrieved from the InterProScan output for statistical analysis. Each Pfam annotation was compared either to the references or between samples within taxonomical groups using fisher exact tests to search for enriched functions. The p-values were corrected for multiple testing using the Benjamini & Hochberg FDR correction. Biological Processes GO terms were analyzed by using the topGO package with the “weight” algorithm. Samples were also compared to each other based on viral Pfam annotations with all viral families together applying Bray-Curtis dissimilarity for clustering.”

The authors’ reply to technical points I concerned before is also not convincing. Authors discussed contamination level of binning based on CheckM results. I agree CheckM partially help to know the contamination level of the MAG. However, the authors must be aware that the tool was developed for bacteria and archaea and no one proved the CheckM results can be equally applied to viruses.

We agree that CheckM does not specifically assess contamination of the viral scaffolds, instead we used it to assess the overall level of confidence we can have in the binning method. In that context CheckM applies as 90% of the data are Bacteria. Independently to CheckM (Fig. S1A), in the previous revision we also performed a dedicated analysis of missassemblies using rich Mock communities (Fig. S1B). This confirms that we can trust the scaffolds with only 0.2% of potential chimeras in this control dataset.

We also used ViralRecall, a tool that is specifically dedicated to check for virus-cell chimeras. The manual check of the viral scaffolds ended up in no cellular-virus chimeras detected. Furthermore, for the large genome fragments we also manually analyzed the coverage with again no sign of missassemblies. Altogether this supports the quality of our assembly. Obviously, working with metagenomic data we cannot guarantee that all the viral scaffolds are correctly assembled.

Furthermore, the authors described their alternative validation is similar to previously published one, named “Jorg”. However, the method in the manuscript is fundamentally different from Jorg. Jorg aims to circularization of the genome, as circularity is one measurement of the bacterial genome completeness. Jorg takes only circularized genomes are considered as true assembly. In contrast, the authors skipped the circularization step, the crucial one for the Jorg.

We agree with the reviewer that the Jorg method is different from ours regarding the circularization step. This step cannot be applied in our case as most of the Nucleocytoviricota genomes are linear. We initially cited the Jorg to indicate that there were other methods that used binning as a way to decomplexify the reads dataset rather than accepting them as complete genomes. In addition, our aim was not to get complete genomes but rather to get an overall view of the viral diversity of our ancient and modern samples. We removed the citation to Jorg in the revised manuscript to avoid any confusion.

In addition, Jorg utilizes another method, mirabait and MIRA, for second mapping and assembly, respectively. Jorg’s paper describes this mapping-based second step is more accurate than k-mer-based method, like SPAdes. The authors did not consider these points and only used SPAdes for first and second assemblies. Collectively, the authors statements did not provide sufficient evidence to support the quality of their assembly.

To date, there is no consensus in the literature as whether MIRA performs better than SPAdes at assembling viral genomes. Some authors indeed indicate better performances (<https://www.liebertpub.com/doi/full/10.1089/cmb.2017.0008>) but others find the opposite (see <https://microbiomejournal.biomedcentral.com/articles/10.1186/s40168-019-0626-5> and <https://www.frontiersin.org/articles/10.3389/fbioe.2015.00141/full>). We nevertheless tested Jorg on the Mock dataset without the circularization step, as again it cannot apply to our study. This is summarized in the table below. Jorg indeed lowers the proportion of potential chimeras but the gain is negligible as our method already performed well (with only 0.2%). Above all, this gain would be at the cost of a loss of real contigs identified with only 4% of the references assembled. Thus, it cannot be used to study diversity, the main purpose of our work.

	Contigs > 5 kb	N50	Chimeric contigs	References found	References
Our method	9190	38115	0.2 % (20)	11 % (440)	3734
Jorg	4187	27135	0 % (1)	4 % (145)	3734

Below are additional major concerns that I did not point before.

Line 228: They insisted they revealed the diversity of the pithoviruses. However, they used contigs and not MAGs in their phylogenetic analysis. In the figure 4, contigs with one marker gene are included. In their method, split genomes cannot be rescued if first binning failed. Therefore, it is highly possible they originated from limited number of genomes, and the pithoviruses diversity was overrepresented.

The reviewer probably refers to the phylogenetic tree in Figure 3. This is a valid point as accepting scaffolds that have missing genes in the tree might introduce a bias with a single genome potentially present several times. Although, there is no reason why Pithoviridae would be more prone to this bias than other viral families. To circumvent this potential bias, we now also provide phylogenies of the individual seven core maker genes (new Fig S8) that are mostly in single copy within genomes (the outlier being the MCP) and thus prevent from artificially inflating the diversity. The figure clearly shows that Pithoviridae are diverse using each of these markers, except for the A32 ATPase and MCP for obvious reasons: the A32 ATPase is missing and the MCP is highly divergent in Pithoviridae. Thus, the combination of both approaches (Fig. 3 and Fig. S8) confirms that Pithoviridae are especially diverse in this type of environment.

Line 333: “patchwork of functions” is not surprising for NCLDVs. This part should be more carefully examined if there are statistically unique functions found in permafrost NCLDVs, using GO and KEGG.

Alongside the comparative analysis of functions enriched within the permafrost samples mentioned above, we also tested for potential enrichment of functions in permafrost samples compared to reference genomes using Pfam and GO terms annotations (Table S7). This analysis highlights the paucity of permafrost-specific Nucleocytoviricota functions with the exception of a GO term related to transcription and the DNA polymerase Pfam domain. This gives nothing conclusive about understanding the permafrost ecology. The patchwork of functions together with very large amount of ORFans makes it hard again to find significantly enriched functions.

We mention this analysis in the result section:

“We searched for significantly enriched Pfam and Gene Ontology annotations in the permafrost viral datasets compared to references but found none after false discovery p-value correction apart from a couple of core function (Table S7).”

And in the discussion:

“The Nucleocytoviricota diversity explored in this study revealed large genomic sequences of unknown families, such as the Permafrost:M_b2437_k1 scaffold (Fig. 4). In addition, we identified many divergent Pithoviridae-like sequences which constitute new clades within the Pimascovirales. In contrast, Megamimivirinae, Klosneuvirinae and Mesomimivirinae were the groups with the least ORFans within the permafrost sequences. These groups are thus better sampled than all other viral families found in this study (Fig. S7A). Overall, the high ORFan content in our dataset probably explains the paucity of functions significantly enriched in permafrost samples compared to reference sequences. In addition, the patchwork-like pattern of Nucleocytoviricota functions might also blur the statistical signal.”

Line 345: The authors did not provide reasonable information why they focus on histones. Thus, results are not related to the permafrost environment. This is also true for translation-related genes.

Permafrost Nucleocytoviricota are indeed not specifically enriched in histones compared to viruses found in aquatic environments. However, those are important functions when studying giant viruses as demonstrated by the very recent studies on virally-encoded histones (see for instance <http://dx.doi.org/10.1038/s41594-021-00585-7> and <https://doi.org/10.1016/j.cell.2021.06.032>). As there were few metagenomic studies on viral histones at the time we submitted this manuscript we took benefit of our metagenomic datasets to address this topic. In the mean time of the reviewing process, a detailed study on virally-encoded histones, including from a metagenomic perspective, has been published (<http://dx.doi.org/10.1186/s13072-022-00454-7>). This lowers the impact of this part of our analysis, we thus decided to remove it from this revised version. Regarding the virally-encoded translation-related genes our work shows inter-viral families' gene exchanges. We believe this is a very important point to understand Nucleocytoviricota evolution that has not been extensively addressed yet. Thus, although it is not specifically linked to permafrost, we believe that this part should remain in the manuscript.

Reviewers' Comments:

Reviewer #3:

Remarks to the Author:

Previously, I pointed their weak connection to the uniqueness of permafrost samples. Judging from the authors' reply, quality of their samples was not enough to connect the samples' uniqueness to their results. Without permafrost-specific insights, the paper is similar to those already done (Schulz et al., 2018, Nature communication; Backstrom et al., 2019, mBio; Schulz et al., 2020, Nature).

The author's reply on their methods is not convincing. Their claims are all indirect evidence for the quality of viral MAGs. They used CheckM and Viralrecall, which can examine prokaryote-prokaryote and cellular-virus contamination, respectively, but virus-virus contamination cannot be investigated. In the reply letter, they mentioned it a little, but it is not clear what was done. Current binning method cannot be perfect, so binning-based assembly should be evaluated carefully. For example, Jorg software uses circularity as an evidence supporting the quality. The authors lack such supporting evidence, and the quality of viral MAGs were not directly evaluated.

Reviewer #4:

Remarks to the Author:

Rigou et al. report the metagenomic discovery of a surprisingly large diversity and abundance of giant viruses in permafrost samples. This finding nicely complements previous studies in which thousands of giant viruses were recovered from metagenomic data sets that were primarily from freshwater and marine samples.

The manuscript is well written and the authors put a great effort on improving and evaluating their giant virus sequence data through thorough benchmarking of their approaches (check for chimerism, figure S1, assignment of contigs to viral vs cellular in figure S2 scaffold filtering figure S3, check for endogenous viruses figure S5)

There is currently no gold standard for QC of metagenome derived giant virus sequences but the extensive effort the authors made is laudable and after looking at it carefully I do not see any major issues with their methodology for contig assembly and QC. To improve the contigs, they used an iterative mapping approach, and indeed, some of their contigs (n=8) could be extended beyond 500kb. This is an elegant approach and perfectly suitable to improve contig length.

I was surprised the authors used CheckM to QC their sequences as this tool was developed for bacterial and archaeal genomes and not for giant viruses. In a recent study (PMID: 32576649) CheckM was run on giant virus reference genomes and most were classified as Archaea. However, CheckM can be used with a custom set of HMMs which would likely improve its performance on giant virus metagenome assembled genomes.

It seems that the authors did not trust metagenomic binning and instead preferred to perform their analysis on the contig level. Analyzing the data on the contig level is a valid approach widely used in most viral metagenomic studies that do not focus on giant viruses, however, as brought up by reviewer 3 this might inflate diversity to some extent - especially due to the fact that there are no reliable single copy genes in giant viruses and many of the "low-copy" marker genes can sometimes unexpectedly be present in many copies (paralogs), potentially spread out over several distinct contigs. Metagenomic binning would most of the time unite these contigs in a single metagenome assembled genome representing a viral population. Binning followed by read mapping may then help to assess strain heterogeneity. I agree with the concern of reviewer 3 that the increase in diversity of the Nucleocytoviricota (and Pithoviruses in particular) shown in figure 3 and discussed in the paper might be inflated. It would be important to have an inclusion criterion for metagenomic contigs into

the species tree based on the presence of a certain number of marker genes. In the case of the 7 GVOGs suitable to be used for the tree (PMID: 34705818) a cutoff of 3 out of 7 would be a good choice. Many contigs that represent partial viral genomes would be removed and the resulting species tree would rather under-represent than over-represent giant virus diversity in the samples as it would show only contigs that represent a substantial proportion of the viral genome. To account for that the figure could be complemented with a new panel that contains summary stats / counts derived from single marker phylogenies shown in Figure S8 (e.g. a box plot for each of the 7 markers on x-axis with data points representing contigs colored by taxonomy and y-axis contig size, and a set of bars that indicate total counts per marker). Ultimately the use of metagenomic binning (instead of using single contigs), as done in many microbial (and giant virus) metagenomic studies, would have helped to improve genome recovery and correctly add more new giant viruses to the tree.

REVIEWER COMMENTS

The reviewers' comments are noted in black font, our responses are in red and citations to the manuscript in green.

Reviewer #3 (Remarks to the Author):

R3.1

Previously, I pointed their weak connection to the uniqueness of permafrost samples. Judging from the authors' reply, quality of their samples was not enough to connect the samples' uniqueness to their results. Without permafrost-specific insights, the paper is similar to those already done (Schulz et al., 2018, Nature communication; Backstrom et al., 2019, mBio; Schulz et al., 2020, Nature).

We disagree with this remark. We believe that our work highlights the abundance and diversity of giant viruses, particularly *Pithoviridae*, in Permafrost. The references mentioned by the reviewer address giant viruses' diversity in mainly freshwater and marine samples. As confirmed by the reviewers 1,2 and 4, this work thus nicely complements the cited references by exploring an untapped environment.

R3.2

The author's reply on their methods is not convincing. Their claims are all indirect evidence for the quality of viral MAGs. They used CheckM and Viralrecall, which can examine prokaryote-prokaryote and cellular-virus contamination, respectively, but virus-virus contamination cannot be investigated. In the reply letter, they mentioned it a little, but it is not clear what was done. Current binning method cannot be perfect, so binning-based assembly should be evaluated carefully. For example, Jorg software uses circularity as an evidence supporting the quality. The authors lack such supporting evidence, and the quality of viral MAGs were not directly evaluated.

There is currently no standard quality check procedure for giant viruses metagenomic assemblies mainly because they are very divergent, mostly composed of ORFans and the very few marker genes are sometimes in multiple copies. This is the reason why we set up several procedures to check the validity of our datasets which includes check for chimerism as mentioned (Fig. S1), viral-cellular contigs classification (Fig. 1, Fig. S2), potential virus endogenization (Fig. S5), as acknowledged by reviewer #4 (cf. R4.1). We also now provide a check for virus-virus contamination using CheckM fueled with virus-specific HMMs (cf. answer to review 4, R4.3).

Reviewer #4 (Remarks to the Author):

R4.1

Rigou et al. report the metagenomic discovery of a surprisingly large diversity and abundance of giant viruses in permafrost samples. This finding nicely complements previous studies in which thousands of giant viruses were recovered from metagenomic data sets that were primarily from freshwater and marine samples.

The manuscript is well written and the authors put a great effort on improving and evaluating their giant virus sequence data through thorough benchmarking of their approaches (check for chimerism, figure S1,

assignment of contigs to viral vs cellular in figure S2 scaffold filtering figure S3, check for endogenous viruses figure S5)

We thank the reviewer for recognizing our efforts to provide giant viruses assembled metagenomes as clean as possible.

R4.2

There is currently no gold standard for QC of metagenome derived giant virus sequences but the extensive effort the authors made is laudable and after looking at it carefully I do not see any major issues with their methodology for contig assembly and QC. To improve the contigs, they used an iterative mapping approach, and indeed, some of their contigs (n=8) could be extended beyond 500kb. This is an elegant approach and perfectly suitable to improve contig length.

Again, thanks for this positive comment.

R4.3

I was surprised the authors used CheckM to QC their sequences as this tool was developed for bacterial and archaeal genomes and not for giant viruses. In a recent study (PubMed ID 32576649) CheckM was run on giant virus reference genomes and most were classified as Archaea. However, CheckM can be used with a custom set of HMMs which would likely improve its performance on giant virus metagenome assembled genomes.

The large majority of our full dataset is composed of prokaryotic sequences. Therefore, we reasoned that we could use CheckM for an initial check of the overall validity of our assembly procedure. This was primarily done to validate our choice in either using bins as is or use them to build a second de novo assembly. Based on the CheckM results, indeed focused on the prokaryotic sequences of our dataset, we opted for the second option and then focus on the Nucleocytoviricota fraction. We rephrased this part of the results section to make this point clearer:

“Previous analysis of this dataset (14) showed that prokaryotes are the most abundant (90% of the total coverage). Accordingly, the assembly of the reads (Table S2) predominantly revealed bacterial contigs (mean=94%, sd=7%) according to the Lowest Common Ancestor (LCA) taxonomy based on BLASTP results against RefSeq. The samples with least bacterial contigs (N and R) still contain 80% of those, along with archaeal (10 and 7% respectively), unclassified (5%), viral (2%) and eucaryotic (1.3% and 1.6% respectively) contigs. Owing to the majority of bacterial contigs we reasoned that CheckM (32) could be applied to assess the overall validity of our assembly procedure of the complete dataset. This resulted in clean contigs with very few potential chimeras (0.004%) and no strain level chimera (Fig. S1A). Next, as is custom in metagenomics studies, we performed a binning of the contigs to obtain less fragmented assemblies (33). This revealed potential chimeras (Fig. S1A). We thus chose not to consider bins as unique organisms but instead we used binning as a procedure to decrease complexity in our datasets. More precisely, the reads were first separated according to the bin they belonged to. Next, a second de novo assembly was made within each bin. This resulted in significantly longer scaffolds and a larger total assembly (Table S2) while keeping contamination at a negligible level (on average 0.005% potential chimeras and again none at the strain level, Fig. S1A). Thus, our method significantly gained in reliability by lowering the proportion of

chimeras in comparison to conventional binning, while providing longer assembled sequences compared to standard assemblies.”

Nevertheless, we followed the reviewer suggestion to use CheckM with a custom dataset of HMM to specifically assess giant viruses scaffold quality. More specifically, we used 2204 NCVOGs (reference 44 in the text) that were in low copy (1.1 copy or less on average) in the original database to build HMMs that were fueled to CheckM. In doing so we show that potential virus-virus contaminations are very low with on average 0.0047% contamination and 0.066% strain heterogeneity. This is now mentioned in the results:

“Applying CheckM specifically fueled with viral HMM profiles made from low-copy NCVOGs (44) on this final sequence dataset resulted in virtually no contamination (mean=0.0047%, sd=0.027%) and strain heterogeneity (mean=0.066%, sd=1.8%). “

And the Materials and Methods section:

“CheckM was also applied on a custom set of HMMs made from the NCVOGs database (44) using the “analyze” and “qa” tools. NCVOGs with 1.1 copies or less in average were used to construct the HMM profiles for a low copy NCVOG database.”

R4.4

It seems that the authors did not trust metagenomic binning and instead preferred to perform their analysis on the contig level. Analyzing the data on the contig level is a valid approach widely used in most viral metagenomic studies that do not focus on giant viruses, however, as brought up by reviewer 3 this might inflate diversity to some extent - especially due to the fact that there are no reliable single copy genes in giant viruses and many of the "low-copy" marker genes can sometimes unexpectedly be present in many copies (paralogs), potentially spread out over several distinct contigs. Metagenomic binning would most of the time unite these contigs in a single metagenome assembled genome representing a viral population. Binning followed by read mapping may then help to assess strain heterogeneity. I agree with the concern of reviewer 3 that the increase in diversity of the Nucleocytoviricota (and Pithoviruses in particular) shown in figure 3 and discussed in the paper might be inflated. It would be important to have an inclusion criterion for metagenomic contigs into the species tree based on the presence of a certain number of marker genes. In the case of the 7 GVOGs suitable to be used for the tree (PubMed ID 34705818 SMASH) a cutoff of 3 out of 7 would be a good choice. Many contigs that represent partial viral genomes would be removed and the resulting species tree would rather under-represent than over-represent giant virus diversity in the samples as it would show only contigs that represent a substantial proportion of the viral genome. To account for that the figure could be complemented with a new panel that contains summary stats / counts derived from single marker phylogenies shown in Figure S8 (e.g. a box plot for each of the 7 markers on x-axis with data points representing contigs colored by taxonomy and y-axis contig size, and a set of bars that indicate total counts per marker). Ultimately the use of metagenomic binning (instead of using single contigs), as done in many microbial (and giant virus) metagenomic studies, would have helped to improve genome recovery and correctly add more new giant viruses to the tree.

We agree with this point. For this reason, we provided a phylogeny of each individual marker gene (now Fig. S9) showing that *Pithoviridae* are the most diverse in permafrost samples in each case (except for the ATPase that is lacking in *Pithoviridae* and the MCP that is highly divergent).

To strengthen this point we followed the reviewer's suggestion and build a phylogeny with scaffolds that contain at least 3 out of the 7 markers. This obviously drastically lowered the sample size (with 37 classified sequences as compared 369 when only one marker gene is required) but still shows the same taxonomy pattern. This phylogeny is presented in the new Fig. 3 in the main text, along with a second panel that contains summary stats/counts as suggested. The former Fig. 3 (including sequences with ≥ 1 marker) is now in supplements (now Fig. S7).

In addition, we followed the reviewer's advice to work on raw bins to examine the phylogeny. This probably noisier dataset covers almost the complete Nucleocytoviricota sequence dataset (85.4%). Again, the dominance of *Pithoviridae* can be observed (see new Fig. S10). We also provide the raw bin sequences as supplements.

To summarize, the analysis of scaffolds that contain ≥ 3 markers (Fig. 3), the ones that contain ≥ 1 marker (Fig. S7), the bins (Fig. S10), the individual markers phylogenies (Fig. S9) and the best BLASTP matches of unclassified sequences (Fig. S8) all converge to the fact that *Pithoviridae* and *Orpheoviridae* families are the most diverse in our permafrost samples.

We modified the main text to account for these additional analyses:

“To further investigate which viral families were present in the samples, we conducted a phylogenetic analysis based on 7 marker genes (Table S3) and a curated database produced by a former study (39). We excluded the transcription elongation factor TFIIS as its phylogeny breaks well established clades (*Alphairidovirinae*, *Ascoviridae*, *Asfarviridae*, *Pimascovirales*, Fig. S6). It should also be noted that the primase D5 revealed an unexpected grouping of the *Cedratviruses* with *Phycodnaviridae* instead of *Pithoviridae*, suggesting that this gene was acquired from an unknown source in *Cedratviruses* (Fig. S6). We first classified permafrost scaffolds containing at least three of the seven marker genes to avoid split genomes in the tree. This resulted in 37 classified scaffolds (corresponding to 16.5% of the 72 Mb of total Nucleocytoviricota identified sequences) with 21 scaffolds within the *Pithoviridae* and *Orpheoviridae*-like clades, 8 in the *Megamimiviridae* clade and the rest associated to *Klosneuviridae*, *Phycodnaviridae* and one *Asfarviridae* (Fig. 3A).

However, filtering scaffolds with less than 3 marker genes only reveals the ones representing a substantial portion of the viral genome and thus probably under-estimate the true diversity of viral families. Indeed, counts derived from single markers (Fig. 3B) show that *Pithoviridae* and *Orpheoviridae*-like sequences might be particularly under-estimated as they lack the packaging ATPase and contain a highly divergent MCP. In addition, they contain a substantially lower fraction of duplicated marker genes than *Megamimivirinae* and *Klosneuvirinae* (Fig. 3B). We thus also performed a classification of all scaffolds containing at least one marker gene. This increased the taxonomically classified dataset to 369 Nucleocytoviricota scaffolds (40.1% of the Nucleocytoviricota sequences). Again, *Pithoviridae* and *Orpheoviridae*-like viral families were the most diverse, followed by *Mimiviridae* (Fig. S7). In contrast,

Marseilleviridae, Alphairidovirinae, Betairidovirinae and Ascoviridae were completely absent in our samples. Interestingly, unclassified sequences do not encode for more ORFans (ORFs with no similar sequence in the public databases) than classified sequences (Fig. S8A). This suggests that these sequences are not more divergent to known relatives than any other Nucleocytoviricota sequence but remained unclassified due to the lack of the marker genes.

We further confirmed the observed taxonomy pattern from individual marker genes phylogenies (Fig. S9) and best BLASTP matches of the unclassified sequences against the nr database (Fig. S8B). Finally, an alternative phylogeny of the bins (instead of scaffolds) probably noisier but representing 85.4% of the total Nucleocytoviricota sequences confirms the pattern (Fig. S10). Altogether, these results clearly support Pithoviridae and Orpheoviridae-like as the most diverse families in our samples. “

Reviewers' Comments:

Reviewer #3:

Remarks to the Author:

I never deny the importance of the study which expanding the knowledge about environmental giant viruses. However, the study is a complementary work to the previous studies. At the same time, the most part of the study is descriptive. Overall, conceptual advances in this study seem limited.

For technical part, CheckM using virus HMM is a nice idea to check MAG quality. In contrast, the data is a little old and the last update was 2014.

Reviewer #4:

Remarks to the Author:

The authors have addressed all points that I raised to my full satisfaction. The revised phylogenomic analysis is sound and the updated figures look great.

REVIEWER COMMENTS

The reviewers' comments are noted in black font and our responses are in red.

Reviewer #3 (Remarks to the Author):

I never deny the importance of the study which expanding the knowledge about environmental giant viruses. However, the study is a complementary work to the previous studies. At the same time, the most part of the study is descriptive. Overall, conceptual advances in this study seem limited.

For technical part, CheckM using virus HMM is a nice idea to check MAG quality. In contrast, the data is a little old and the last update was 2014.

We thank the reviewer for acknowledging our attempt to expand the knowledge of environmental giant viruses. It is true that, like most metagenomic studies of giant viruses, this study is rather descriptive, but we believe it provides an important foundation for further hypothesis-driven studies.

Reviewer #4 (Remarks to the Author):

The authors have addressed all points that I raised to my full satisfaction. The revised phylogenomic analysis is sound and the updated figures look great.

We thank the reviewer for providing helpful advice in finalizing this manuscript.